# Phosphoproteome profiling uncovers a key role for CDKs in TNF signaling

Maria C. Tanzer [1✉], Isabell Bludau [1], Che A. Stafford[2], Veit Hornung [2] & Matthias Mann [1✉]

Tumor necrosis factor (TNF) is one of the few cytokines successfully targeted by therapies against inflammatory diseases. However, blocking this well studied and pleiotropic ligand can cause dramatic side-effects. Here, we reason that a systems-level proteomic analysis of TNF signaling could dissect its diverse functions and offer a base for developing more targeted therapies. Therefore, we combine phosphoproteomics time course experiments with sub-cellular localization and kinase inhibitor analysis to identify functional modules of protein phosphorylation. The majority of regulated phosphorylation events can be assigned to an upstream kinase by inhibiting master kinases. Spatial proteomics reveals phosphorylation-dependent translocations of hundreds of proteins upon TNF stimulation. Phosphoproteome analysis of TNF-induced apoptosis and necroptosis uncovers a key role for transcriptional cyclin-dependent kinase activity to promote cytokine production and prevent excessive cell death downstream of the TNF signaling receptor. This resource of TNF-induced pathways and sites can be explored at http://tnfviewer.biochem.mpg.de/.

[1] Department of Proteomics and Signal Transduction, Max Planck Institute of Biochemistry, Martinsried 82152, Germany. [2] Gene Center and Department of Biochemistry, Ludwig-Maximilians-Universität, Munich 81377, Germany. ✉email: tanzer@biochem.mpg.de; mmann@biochem.mpg.de

Post-translational modifications (PTMs) such as phosphorylation govern the activation, strength, and timing of immune-signaling pathways. The interplay between kinases and phosphatases results in the rapid addition and removal of phosphates, providing exquisitely precise control of signaling events. In addition to a small number of key phosphorylation sites with switch-like functions, there is a vast number of phosphorylation events that fine-tune cellular responses[1,2].

The TNF pathway is a pivotal proinflammatory signaling cascade that relies heavily on protein phosphorylation, the extent of which has been revealed by phosphoproteomic studies[3–8]. Much is already known about key mechanistic events and we briefly recapitulate those that are pertinent to our study. TNF binds to its receptor (TNFR1), which leads to the recruitment of the adaptor proteins TNF receptor-associated factor 2 (TRAF2), tumor necrosis factor receptor type 1-associated death-domain protein (TRADD), and the receptor-interacting serine/threonine-protein kinase 1 (RIPK1), which play critical roles in the decision between cell death, survival, and inflammation[9,10]. This complex is then ubiquitylated, allowing recruitment and activation of the master kinase TGF-β-activated kinase-1 (TAK1), which then activates MAPK- and NF-κB signaling by phosphorylating several MAPK kinases and IκB kinases (IKK1/2)[11,12]. Both pathways are crucial for the upregulation of many target genes, including a range of cytokines and prosurvival factors[13]. One such prosurvival factor is the cellular FLICE-like inhibitory protein (cFLIP), which is required to inhibit caspase-8 activity and prevent cell death[14]. Disruption of these phosphorylation-driven signaling cascades strongly perturbs gene activation, leading to cell death and inflammation[15]. Besides their roles in transcriptional activation, IKK2 and MAPK-activated protein kinase 2 (MK2) directly phosphorylate RIPK1, inhibiting cell death by altering RIPK1's kinase and adaptor function[16–18]. TNF-induced cell death can either be caspase-dependent (apoptosis) or -independent (necroptosis). Necroptosis induction occurs in the absence or upon inhibition of caspase-8. It involves RIPK1 autophosphorylation and activation of RIPK3, which subsequently phosphorylates and activates the mixed lineage kinase domain like pseudokinase (MLKL), leading to plasma-membrane permeabilization and cell death[19–21]. Necroptosis is considered to be inflammatory[22]. In line with this concept, we previously showed that MLKL activation leads to the release of proteases, and other intracellular proteins[23].

Clinically targeting TNF is highly successful in treating inflammatory pathologies[24–26]. Most anti-inflammatory therapies, including TNF-blocking antibodies, act by inhibiting cytokines that drive inflammation, leading to a complete abrogation of all downstream-signaling events. Considering the pleiotropic functions of cytokines—especially in fighting infection and regulating a range of important signaling pathways—it is not surprising that anti-inflammatory therapies can cause severe side effects[25]. In light of this, targeting specific phosphorylation events by inhibiting certain kinases or phosphatases could allow more precise manipulation of disease pathways while retaining signal transduction required for homeostasis. However, such an approach requires an in-depth knowledge of kinases, phosphatases, substrates, and their signaling dynamics, which is currently unavailable. Furthermore, the complexity and interplay of these events further make analysis by classical cell-biology tools difficult. We therefore reasoned that system-level approaches like proteomics and phosphoproteomics provide the required global cellular perspective on phosphorylation dynamics.

To investigate the TNF-regulated phosphoproteome we combined TNF biology in myeloid-cell systems with a high-sensitivity phosphoenrichment protocol[27,28] and state-of-the-art mass spectrometry. We specifically made use of the data completeness of the data-independent acquisition mode[29]. We employed time course and spatial proteomics to elucidate signaling events downstream of the TNF receptor. Based on this in-depth data, we functionally probed the impact of master kinases and TNF-induced cell death on the global TNF phosphoproteome, revealing a plethora of signaling events not previously connected to TNF. Our findings offer a comprehensive resource of phosphorylation events regulated upon TNF stimulation and TNF-induced cell death, which is available to the community at http://tnfviewer.biochem.mpg.de/. We provide evidence for TNF-mediated cross-talk with other innate immune-signaling pathways and identify a role for CDK-kinase activity in TNF-induced cell death.

## Results

**Temporal analysis of TNF-induced phosphorylation events enables separation of early and late signaling events.** To acquire a system-based view of phosphorylation events and their kinetics downstream of the TNF-receptor complex, we treated the myeloid cell line U937 with TNF over a time course of fifteen seconds to one hour (Fig. 1a, b). We analyzed phosphopeptides in data-independent acquisition (DIA) and detected more than 60,000 phosphopeptides across several experiments (Supplementary Fig. 1a). In total, 28,000 class-I phosphosites (localization probability > 75%) (Fig. 1c) were detected in the time-course experiment, which was measured with high reproducibility (Supplementary Fig. 1b). Significantly changing TNF-induced phosphosites peaked at 15 min, but we detected significant upregulations already at 3 m after treatment (Fig. 1d, e). The large majority returned to baseline by 60 min (Fig. 1e, f). Many of these transiently modified proteins are involved in NF-κB- and pattern-recognition signaling. In contrast, a small cluster of phosphosites remains upregulated even at 60 min post TNF stimulation and were located on proteins involved in transcription (Fig. 1f). We categorized phosphosites into early (significantly regulated up to 5 min of stimulation), middle (at 15 min), and late (at 60 min) events (Supplementary Fig. 1c–e). Fisher's exact test on the different temporal sections revealed the dynamic regulation of the respective cellular processes. Within five minutes of TNF-stimulation, the phosphorylation status of proteins involved in regulation of vesicle fusion and myeloid-cell differentiation was increased, while terms involving the RIG-I-, NF-κB -, TRIF- and MYD88 pathways, as well as GTPase activation, followed at 15 min (Supplementary Fig. 1d). Terms related to transcription were regulated throughout the time course—most strongly at the latest time point, consistent with transcription being the most downstream process of the TNF-signaling cascade. Motif-enrichment analysis revealed a dynamic activation of different kinases along the time course (see "Methods", Supplementary Fig. 1f). *CDK1/2* and *PLK1/3* motifs were downregulated, whereas *IKBKB* (IKK2) and *PRKAA1/2* motifs were upregulated at early time points. Phosphorylations on one protein could present with different kinetics. For example, of the quantified phosphosites of INPP5D, a phosphatase involved in immune signaling, S886 and T1108 peaked at 8 min, whereas T963 and T971 only peaked at 60 min, which suggests that different regulators act on one protein (Supplementary Fig. 1g).

With the time course on U937 cells in hand, we selected the 15-min time point to interrogate TNF signaling in other cell lines (A549 (adenocarcinomic alveolar basal epithelial cell line), HT29 (colorectal adenocarcinoma cell line), and U2OS (osteosarcoma cell line)) and another relevant primary cell system, murine bone marrow-derived macrophages (BMDMs). Many phosphorylation events significantly regulated upon TNF treatment are shared across the different human cell lines (Supplementary Data 1).

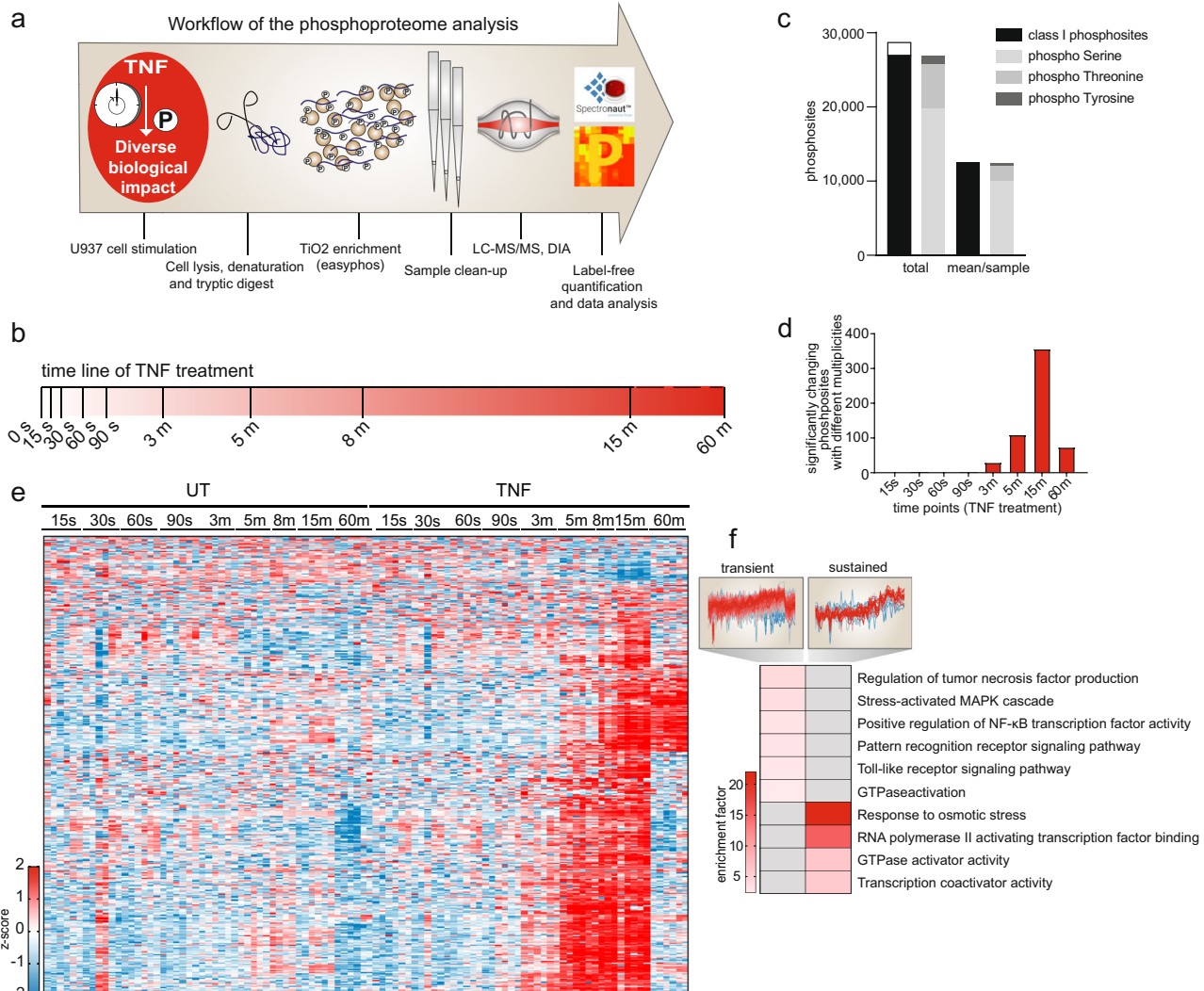

**Fig. 1 Temporal dissection of TNF-regulated phosphorylation events. a** Workflow of the phosphoproteome analysis. **b** Schematic overview of the time points used for TNF treatment (100 ng/ml). **c** Bar graph demonstrating the number of detected phosphosites in U937 cells. Class I sites (in black) are phosphosites with a localization probability of > 75%. **d** Histogram depicting the numbers of significantly changing phosphosites along eight different time points compared with the respective untreated control (FDR < 0.05; see Methods for an 8-min time point). **e** Heatmap of significantly changing phosphosites along the time course compared with their respective untreated controls (Student's t-test; FDR < 0.05). Z-scores of various time points and the replicates (n = 4 biologically independent experiments) in red for intensities higher than the mean and in blue for intensities lower than the mean of the respective phosphosites across all samples. **f** Fisher's exact test of transient phosphosites induced upon TNF, which are downregulated after 15 min and sustained phosphosites, which are still upregulated at 60 min of treatment (two-sided, p < 0.002). The red scale resembles enrichment, while gray represents missing values/no enrichment. The profiles are color-coded according to their distance from the respective cluster center (red is close to the center, blue is further away from the center). Source data are provided as source file.

This is also reflected by the same enrichment terms resulting from Fisher's exact test of significantly upregulated phosphosites between the different cell systems (Supplementary Fig. 1h, i). While we observed classical TNF-signaling phosphorylations, surprisingly, many proteins involved in other immune-response pathways like MAVS (S222, S419) in U937 cells or NLRC4 (S533) and OAS3 (S384) in BMDMs were also dynamically phosphorylated (Supplementary Fig. 1j–l, m; Supplementary Data 2). Terms like peptidoglycan response, TLR signaling, and response to exogenous dsRNA were significantly enriched, suggesting a cross-priming function for TNF upon infection (Fisher's exact test Supplementary Fig. 1l, m).

**Kinase-hub inhibition unravels kinase-substrate relations upon TNF stimulation**. While our analysis of phosphorylation dynamics enabled the temporal dissection of phosphorylation events, identification of specific kinase-substrate relationships remained challenging due to the simultaneous activation of many downstream kinases. To address this, we targeted key kinases such as TAK1 using specific inhibitors (Supplementary Fig. 2a). TAK1 plays a crucial, upstream role in activating NF-κB signaling by phosphorylating IKKs (Fig. 2a). This pathway is required for the upregulation of prosurvival target genes and is thereby essential for cell survival upon TNF stimulation (Fig. 2b, Supplementary Fig. 2b). TAK1 is also important for the activation of the MAPK pathway by phosphorylating MAPKs like p38, MEK, and JNK (Fig. 2a). While TNF stimulation alone induced the regulation of hundreds of phosphorylation events, inhibiting TAK1 almost completely abrogated its effect (Fig. 2c, d; Supplementary Fig. 2c). This confirms the upstream role of TAK1 in TNF signaling using a global phosphoproteomics readout.

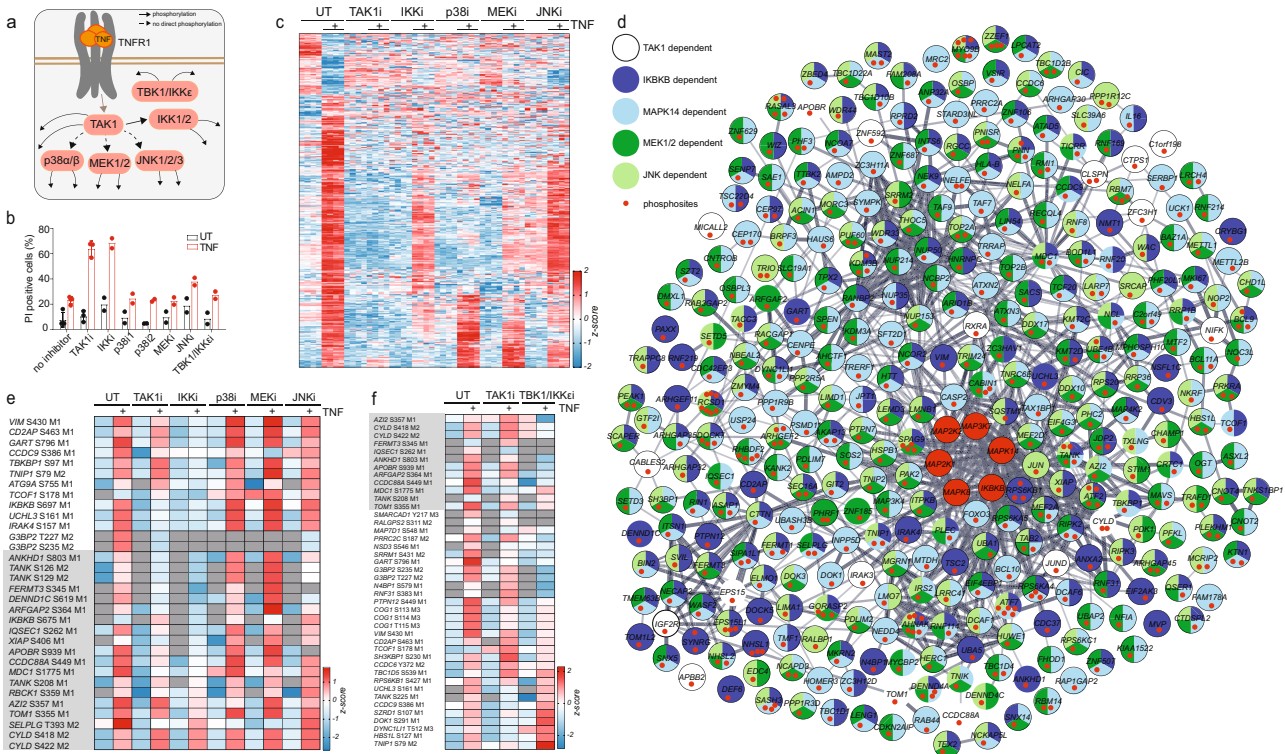

**Fig. 2 Kinase inhibition unravels kinase-substrate relations upon TNF stimulation. a** TNF-signaling scheme highlighting kinase hubs that we inhibited in this study. **b** Cell-death analysis by flow cytometry of propidium iodide-stained U937 cells either stimulated with the indicated kinase inhibitors alone (TAK1 inhibitor (7-Oxozeaenol, 1 μM), IKK inhibitor (TCPA-1, 5 μM), p38 inhibitors 1 and 2 (LY228820, 2 μM; SB203580, 10 μM), MEK inhibitor (PD025901, 10 μM), and JNK inhibitor (SP600125, 20 μM)) or pretreated with kinase inhibitors 1 h before TNF treatment for 24 h (±SD, $n = 2$ biologically independent experiments). **c** Heatmap of z-scored phosphosite intensities significantly changing in U937 cells treated for 15 min with TNF compared with untreated cells (two-sided Student's $t$-test FDR < 0.05). Cells were additionally pretreated for 1 h with kinase inhibitors as indicated in (**b**). Z-scores of the four replicates are shown. **d** Network analysis (STRING) of significantly upregulated phosphopeptides upon TNF treatment (two-sided Student's $t$-test -Log10 $p$ > 1.3, Log2 fold change > 1.3). Phosphopeptides that were significantly increased only upon inhibitor treatment were excluded (two-sided Student's $t$-test −Log10 $p$ > 1.3, Log2 fold change > 0). Phosphopeptides not significantly increased upon TNF and inhibitor treatment were considered inhibited (two-sided Student's $t$-test -Log10 $p$ < 1.3 and/or Log2 fold change < 1.3). **e** Heatmap of means of z-scored phosphosite intensities that are induced upon TNF stimulation, despite TAK1 inhibition (two-sided Student's $t$-test of cells treated with TAK1 inhibitor alone compared with TAK1 inhibitor with TNF, -Log10 $p$ > 1.3, Log2 fold change > 1.3 using two different imputations, Methods). Gray represents missing values. Phosphosites with gray background are not ablated by any inhibitor used. **f** Heatmap of means of z-scored phosphosite intensities that are induced upon TNF stimulation despite TAK1 inhibition (two-sided Student's $t$-test of cells treated with TAK1 inhibitor alone compared with TAK1 inhibitor with TNF, -Log10 $p$ > 1.3, Log2 fold change > 1.3 using two different imputations). Cells were treated with: TAK1 inhibitor (7-Oxozeaenol, 1 μM) and TBK1/IKKε inhibitor (MRT67307, 2 μM). Gray represents missing values. Phosphosites with gray background were not ablated by any inhibitor used in the experiment of (**e**). Source data are provided as source file.

In contrast, inhibition of downstream kinases such as IKK1/2, p38, MEK1/2, and JNK had a modest-to-intermediate effect on TNF-regulated phosphosites (Fig. 2c; Supplementary Fig. 2c). Inhibiting p38 had the strongest impact, reducing 76% of the TNF-upregulated phosphosites (Supplementary Fig. 2c). Proteins phosphorylated upon TNF stimulation that have not been associated with other phosphorylated proteins in the STRING network, can now be assigned to upstream kinases (Fig. 2d).

These experiments also identified upstream kinases responsible for phosphorylation events on classical TNF-signaling complex members (Supplementary Fig. 2d). Phosphorylation of the TRAF2 and NCK-interacting protein kinase (TNIK) (S640) and of Mind bomb-2 (MIB2) (S309), a newly discovered E3 ligase in TNF signaling[30], was inhibited by TAK1, p38, and MEK inhibition. XIAP and TRAF2 phosphorylation was strongly reduced by TAK1 and IKK2 inhibition. A recent phosphoproteomics study reported that the phosphorylation of Metadherin (MTDH) at position S298 is important for NF-κB signaling and is also mediated by IKK2[6]. However, we failed to detect down-regulation of this site due to the IKK1/2 inhibitor, but instead

measured inhibition with two different p38 inhibitors in two independent experiments (Supplementary Fig. 2e). Surprisingly, phosphorylation sites on TNFAIP3-interacting protein 1 (TNIP1), TANK-binding kinase 1-binding protein 1 (TBKBP1), autophagy-related protein 9 A (ATG9A), and other proteins were not abrogated by TAK1 inhibition but were strongly affected by IKK1/2 inhibition (Fig. 2e). This demonstrates that IKK1/2 must retain some function independent of TAK1 activity.

Furthermore, we detected two novel phosphorylation sites (T136 and S359) on HOIL1, also called *RBCK1* for RanBP-type and C3HC4-type zinc finger-containing protein 1, an essential member of the linear ubiquitin-assembly complex (LUBAC) (Supplementary Fig. 2d). The phosphorylation at T136 was inhibited by blocking TAK1 and IKK2 activity, while phosphorylation at S359 did not change significantly. Phosphorylation sites on other proteins associated with LUBAC, including the ubiquitin carboxyl-terminal hydrolase (CYLD) and the 5-azacytidine-induced protein 2 (AZI2), also occurred independently of the kinase inhibitors used above (Fig. 2e). LUBAC is required for the recruitment and activation of TANK-binding kinase 1 (TBK1)/

inhibitor of nuclear factor kappa-B kinase subunit epsilon (IKKε), and IKKε phosphorylates and inhibits the deubiquitinase CYLD[31,32]. Therefore, we tested whether TBK1 and IKKε are responsible for phosphorylating other LUBAC members and associated proteins. Indeed, inhibiting TBK1/IKKε affected not only TNF-induced phosphorylation of CYLD but most of the other proteins, whereas TAK1 inhibition failed to block these phosphorylation events, including on ADP-ribosylation factor GTPase activating protein 2 (*ARFGAP2*), IQ motif and SEC7-domain-containing protein 1 (*IQSEC1*), coiled-coil domain containing 88b (*CCDC88A*), and fermitin family homolog 3 (*FERMT3*) (Fig. 2f). Our data thus suggest a potential association of these proteins with the LUBAC complex upon TNF signaling.

**TNF-mediated phosphorylation induces widespread protein relocalization.** TNF-dependent phosphorylation relies on the correct localization of kinases and their substrates and requires subcellular protein translocation[33]. TNF binding to its receptor recruits a range of complex members to the plasma membrane and binding of TRAF2, and requirement of TRADD, RIPK1, IAPs, and the LUBAC complex for proper activation of TAK1 is well documented[34]. However, the impact of phosphorylation on protein localization is less explored. We therefore set out to obtain a system-based view on protein translocations and the role of phosphorylation on these translocations upon TNF stimulation. We treated cells with TNF for 15 min with and without the TAK1 inhibitor and subsequently separated membrane, nucleus, and cytosol (Fig. 3a). Peptides were either phosphoenriched or directly measured for full-proteome analysis. Compartment markers were strongly enriched in their respective fraction, indicating successful fractionation (Supplementary Fig. 3a). Most measured proteins and phosphopeptides were significantly enriched in one of the indicated fractions compared with other fractions (Fig. 3b, c). We detected TRADD and RIPK1 at the plasma membrane after stimulation, providing a positive control of our spatial proteomics experiment (Fig. 3d). Their recruitment is TAK1 independent, confirming their upstream role in TNF signaling. NF-κB essential modulator (NEMO or *IKBKG*), TRAF2 and TNIP1 levels also increased in the membrane fraction upon TNF stimulation (Fig. 3d). While S77, S82, and S79 on TNIP1 were independent of TAK1 activity (Supplementary Fig. 2d), its recruitment to the membrane was TAK1 dependent (Fig. 3d). Apart from known TNF-complex members, several other proteins also translocated to the membrane fraction. Although this does not prove their direct association with the TNF-receptor complex, it demonstrates their translocation during TNF signaling. Some proteins dissociated from the membrane compartment upon TNF treatment (Supplementary Fig. 3b). The most regulated proteins at the membrane were enriched for proteins involved in the death-domain-mediated complex assembly and the regulation of necrotic cell death ($p < 0.002$; Fisher's exact test; Supplementary Fig. 3c). Surprisingly, we detected increased levels of TAK1 (*MAP3K7*), IKK1 (*CHUK*), and TAB1/2 in the membrane upon TAK1 inhibition (Supplementary Fig. 3d). This may be due to a stabilizing effect of TAK1 inhibitor on TAK1 at the membrane.

Within 15 min of stimulation, TNF triggered nuclear translocation of many proteins involved in transcription, indicating a transcriptional response (Fig. 3e). The transcription factor NFKB1 was the most strongly enriched protein in the nucleus and this was entirely prevented by TAK1 inhibition (Fig. 3e). TAK1 inhibition also increased RIPK1 levels in the nucleus, suggesting that phosphorylation prevents RIPK1 nuclear translocation (Supplementary Fig. 3e). Protein translocation to and from the cytosol is also heavily phosphorylation dependent (Fig. 3f).

Most kinases, including TAK1 (*MAP3K7*), p38 (*MAPK14*), IKK2 (*IKBKB*), and MEK1/2 (*MAP2K1/2*), are primarily enriched in the cytosol, independent of TNF stimulation (Supplementary Fig. 3f). TNF-regulated phosphosites on proteins downstream of these kinases are, however, enriched in all cellular compartments (Fig. 3g). For example, in the nucleus, we detected TNF-regulated phosphosites on proteins involved in RNA polymerase-II transcription (Fig. 3h). Proteins of the RIG-I pathway and cytoplasmic RNA-processing body assembly are differentially phosphorylated in the membrane, and phosphosites within pattern-recognition receptor signaling and cell-migration pathways are regulated in the cytosol (Fig. 3h). Most of these are dependent on the activity of TAK1, which is localized in the cytosol, implying that TNF triggers protein translocation of kinases and their substrates (Fig. 3g). We even measured TNF-regulated phosphorylation events on peptides within fractions where we failed to detect the total protein. This indicates that phosphorylations of these substrates trigger translocation, or that upon translocation, the proteins are phosphorylated immediately (Supplementary Fig. 3g–j).

**TNF-induced cell death triggers a strong RNA-processing response and CDK activation.** Inhibition of the NF-κB-signaling pathway by targeting TAK1 and IKK2 induces cell death downstream of TNF (Fig. 2b)[15]. This cell death can either be apoptotic (caspases active) or necroptotic (caspases inactive). To dissect the role of phosphorylation events regulated upon cell death, we treated U937 cells for three hours with TNF alone, or in combination with the cIAP inhibitor Smac mimetic (Birinapant) to induce apoptosis, or with Smac mimetic and the caspase inhibitor Idun (IDN-6556) to induce necroptosis (Fig. 4a). We also performed experiments in BMDM cells that were pretreated with TAK1 and caspase inhibitors prior to TNF stimulation (Supplementary Fig. 4a). This identified several phosphorylations on members of the TNF-signaling pathway (Fig. 4b). The classical activating phosphorylations S418/S422 on CYLD were reduced, more likely due to CYLD cleavage by active caspases than active dephosphorylation[35] (Fig. 4b). S5, S406, and S430 on XIAP, an E3 ligase known to inhibit TNF-induced cell death, were also strongly regulated upon caspase activation (Fig. 4b). S430 induces XIAP autoubiquitination, resulting in its degradation, leading to increased cell death upon viral infection[36]. However, when we tested the impact of the phosphomimetic and -ablating XIAP mutants on TNF-induced cell death, we failed to detect differences compared with cells expressing XIAP wildtype (Supplementary Fig. 4b–d). This does not exclude potential functionality for these phosphorylation events in a different cell system upon different stimuli. In BMDMs, we detected RIPK3 phosphosites that were upregulated during necroptosis (Supplementary Fig. 4e, f). S14 and S25 of RIPK1 similarly increased during cell death, whereas S15, S313, S321, and S415 did not. With respect to major cellular processes regulated during apoptosis, we found that phosphosites on proteins involved in RNA splicing and mRNA processing significantly upregulated in both cell types (344 phosphosites in U937 cells and 123 phosphosites in BMDMs, FDR < 0.05, Student's $t$ test) (Fig. 4c; Supplementary Fig. 4g–i). These processes were not only induced upon caspase activation but to a smaller degree also during necroptosis (Fig. 4c; Supplementary Fig. 4h). In contrast, the activation of the DNA damage sensor kinase ataxia telangiectasia mutated (ATM) upon TNF-induced cell death was strictly dependent on caspase activation. Indeed, DNA fragmentation via the caspase-activated DNase triggers this DNA damage response upon TNF-induced apoptosis[37] (Fig. 4c; Supplementary Fig. 4g). Necroptosis induced the phosphorylation of membrane proteins, most likely as a consequence of

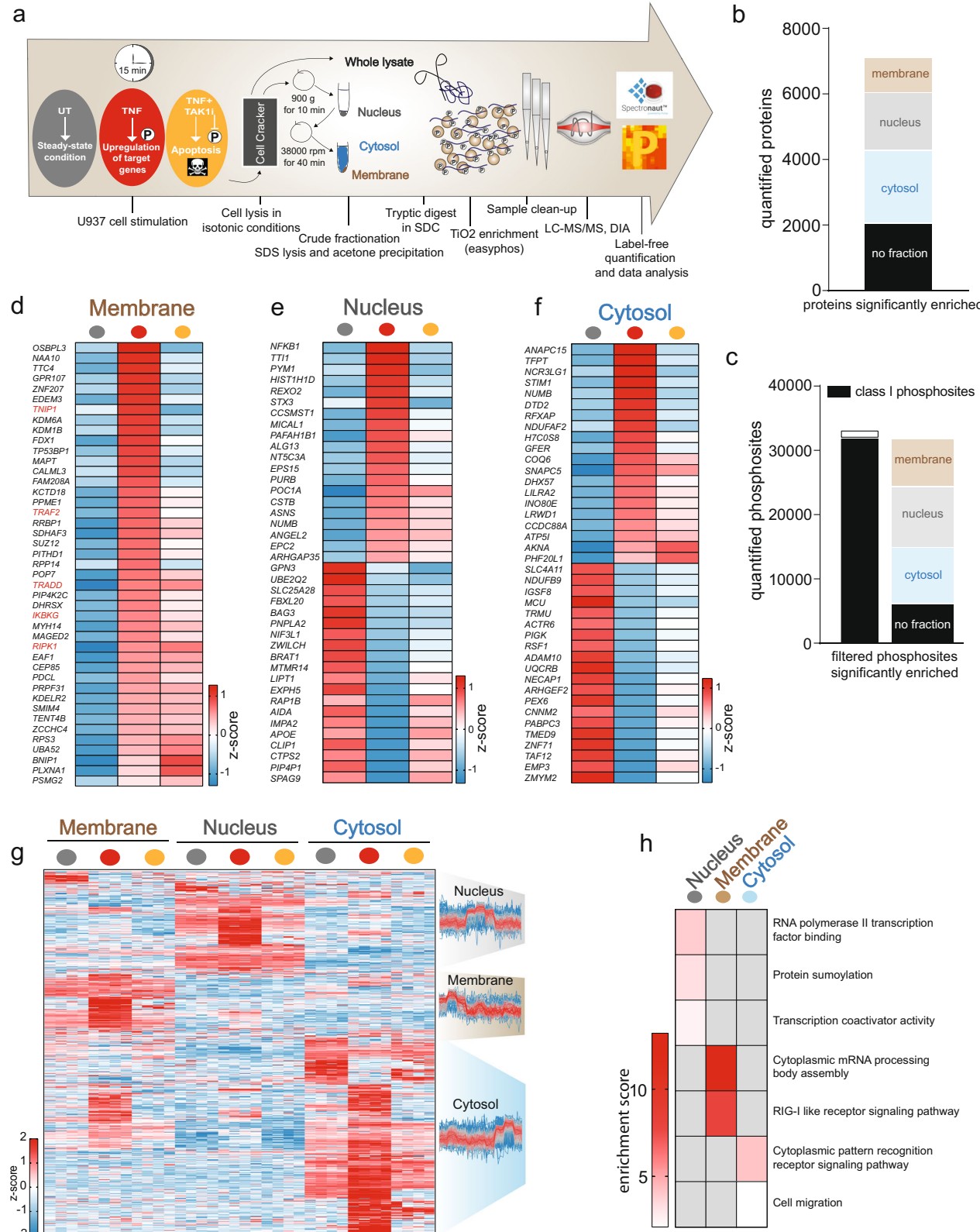

plasma-membrane permeabilization in U937 cells, and it caused a strong GTPase response in BMDMs (Fig. 4c; Supplementary Fig. 4g). Proteome analyses in both cell lines revealed that stimulation-induced phosphoproteome changes did not correlate with proteome changes (Supplementary Fig. 4j–m).

Phosphorylation of kinase motifs in CDK1, 4, 5, and 6 in apoptotic and necroptotic cells was downregulated, suggesting inhibition of the cell cycle during cell death (Fig. 4c; Supplementary Fig. 4n). Enrichment analysis on kinases whose phosphorylation was regulated upon stimulation revealed regulation of cyclin-dependent protein kinases during apoptosis (Fig. 4d). Filtering for ANOVA significantly changing phosphosites on CDKs upon stimulation, showed that transcriptional CDKs (CDK7, 9, 12, 13, and 14) were regulated rather than CDKs

**Fig. 3 TNF-induced phosphorylation strongly affects translocation of signaling members downstream of TAK1. a** Workflow of crude fractionation with subsequent proteome and phosphoproteome analysis of U937 cells. Gray = untreated, red = TNF, orange = TNF + TAK1 inhibitor. **b** Most quantified proteins are significantly enriched in one of membrane, nucleus, and cytosol fractions (two-sided Student's t-test FDR < 0.05). **c** Total number of quantified phosphosites and the numbers of phosphosites significantly enriched in the three fractions (two-sided Student's t-test FDR < 0.05). Class-I sites (in black) are phosphosites with a localization probability of > 75%. **d** Heatmap of means of z-scored protein intensities enriched in the membrane upon TNF treatment (two-sided Student's t-test, -Log10 p > 1.29, Log2 fold change > 0.5). **e** Heatmap of means of the 20 most strongly up- and downregulated z-scored protein intensities in the nucleus upon TNF treatment (two-sided Student's t-test -Log10 p > 1.5, Log2 fold change > 0.5 or < −0.5). **f** Heatmap of means of the 20 most strongly up- and downregulated z-scored protein intensities in the cytosol upon TNF treatment (two-sided Student's t-test -Log10 p > 1.5, Log2 fold change > 3 or < −3). **g** Heatmap of z-scored phosphosite intensities that are significantly regulated upon TNF treatment in the membrane, nucleus, and cytosol (two-sided Student's t-test FDR < 0.05). The profiles are color-coded according to their distance from the respective cluster center (red is close to the center, blue is further away from the center). **h** Fisher's exact test of phosphosites significantly enriched in the nucleus, membrane (p < 0.001), and cytosol (p < 0.01). The red scale represents enrichment, while gray represents no enrichment. Source data are provided as source file.

---

involved in the cell cycle regulation (Methods, Fig. 4e, f). Transcriptional CDKs modulate transcription primarily by regulating RNA polymerase-II activity[38] (Fig. 4d). This occurs by phosphorylation of the carboxyterminal domain (CTD) of its largest subunit RPB1, which consists of 52 heptad sequences (Supplementary Fig. 4o). CDK9 is part of the p-TEFb complex, which induces RPB1-mediated transcriptional elongation by phosphorylating its CTD repeats[39]. Following three hours of TNF treatment, phosphorylation on T354 of CDK9 increased 9-fold (Fig. 4e). This C-terminal site is autophosphorylated by active CDK9[40]. To demonstrate CDK9 activation upon TNF stimulation, we blotted for the CDK9-activating phosphorylation at position T186 and observed a moderate upregulation (Fig. 4g, i). Many phosphorylation sites on both CDK12 and 13 were differentially regulated upon TNF treatment and TNF-induced cell death (Fig. 4e, f). The role of CDK12 and CDK13 in RPB1-mediated transcription was discovered a decade ago, and more recently, CDK12 was shown to be important for intact expression of long genes and reported as a potential antitumor target[41,42]. However, little is known about their regulation via phosphorylation. To test whether these transcriptional CDKs actually phosphorylate RNA polymerase II upon TNF treatment, we blotted for phosphorylation of RPB1 at the CTD heptad sequence position S2. We detected increased phosphorylation of RPB1 upon TNF stimulation, which was reduced by the pan-CDK inhibitor Dinaciclib (Fig. 4g–i). Importantly, THZ531 and AZD4573, specific inhibitors of CDK12/13 and CDK9, respectively[42–44], also inhibited phosphorylation of S2 of RPB1, with AZD4573 exhibiting a more potent effect (Fig. 4h, i). This indicates that TNF signaling induces phosphorylation and thereby CDK9 and CDK12/13-mediated activation of RPB1.

**CDK-kinase activity is required for the transcription of TNF-target genes.** TNF triggers a potent transcriptional response by inducing a wide range of target genes. Its proinflammatory properties are primarily mediated through the upregulation of cytokines, which regulate the immune response, while the upregulation of prosurvival genes prevents excessive cell death and inflammation. Our data demonstrated regulated RPB1 phosphorylation and as this protein is required for TNF-mediated transcription, we wondered if inhibition of transcriptional CDKs would impact TNF-induced target-gene expression. Global phosphoproteomics of CDK inhibitor-treated cells revealed inhibition of TNF-induced phosphorylation of CDK12/13 as well as RNA processing at early and late time points (Supplementary Fig. 5a–c). Interestingly, proteome analysis of TNF-treated U937 cells revealed that Dinaciclib or the two CDK12 inhibitors CDK12-IN3[45] and THZ531 almost completely ablated protein regulation (Fig. 5a). These CDK inhibitors also decreased upregulation of the classical target genes ICAM1 and SOD2, albeit not to the same extent (Fig. 5b). Likewise, there was less upregulation

of OTULIN, a DUB that deubiquitylates LUBAC and thereby prevents cell death[46]. The slight downregulation of XIAP was abrogated by CDK inhibition, which indicates that this regulation is dependent on transcription (Fig. 5b). NF-κB signaling itself—the main pathway driving TNF-induced transcription—was unaffected (Fig. 5c, Supplementary Fig. 5d). To verify that CDK inhibitors indeed inhibit the expression of target genes at the transcriptional level, we examined their impact on TNF-mediated upregulation of *CCL2* (MCP1) and *CXCL10* (IP10) mRNA levels, which were indeed reduced (Fig. 5d, Supplementary Fig. 5e). Similarly, less of the cytokine IP10 was released from BMDMs upon addition of CDK inhibitors (Supplementary Fig. 5f). Consistent with previous studies that investigated the role of CDK9, this indicates that inhibiting transcriptional CDKs 9, 12, and 13 may have an anti-inflammatory effect in disease[47,48]. The induction of the prosurvival proteins A20 and MCL-1 was affected by the pan-CDK inhibitor Dinaciclib but not by the CDK12/13 inhibitor THZ531, while FLIP levels were strongly inhibited by all CDK inhibitors tested (Fig. 5e–h; Supplementary Fig. 5g, h). Pan-CDK and CDK9 inhibition strongly reduced FLIP$_L$ and FLIP$_S$ expression upon TNF stimulation (Fig. 5e, f), while CDK12/13 inhibitors reduced FLIP$_L$ but increased FLIP$_S$ levels compared with TNF stimulation alone (Fig. 5g, h; Supplementary Fig. 5h).

**Transcriptional CDKs inhibit TNF-induced cell death.** FLIP inhibits apoptosis through binding to caspase-8, thereby preventing its activation[49]. FLIP/caspase-8 heterodimers also potently inhibit necroptosis[50]. A previous study presented CDK9 as a potential target for tumor therapy in combination with TRAIL treatment[51]. Therefore, we wondered whether inhibition of TNF-mediated FLIP upregulation by transcriptional CDK inhibitors would enhance TNF-induced cell death. Treating cells with the pan-CDK inhibitor dinaciclib, the CDK9 inhibitors Flavopiridol and AZD4573, the CDK12 inhibitor CDK12-IN3, or the CDK12/13 inhibitors THZ531 and SR4835 all triggered synergistic cell death in combination with TNF, TNF and Smac mimetic (SM), or SM and IDN-6556 (Fig. 6a–d; Supplementary Fig. 6a, b). Inhibitors targeting CDKs that are not involved in transcription failed to induce synergistic cell death, while CDK12 knockdown moderately increased TNF-dependent cell death (Supplementary Fig. 6a, c, d). These experiments show that transcriptional, but not nontranscriptional CDKs exacerbate TNF-induced cell death.

CDK9 and CDK12/13 inhibition, in combination with TNF stimulation, enhanced apoptosis, as evidenced by increased caspase-8 and caspase-3 cleavage (Fig. 6e–g, Supplementary Fig. 6e). This only depended slightly on RIPK1-kinase activity demonstrated by necrostatin-1 treatment (Fig. 6a–d; Supplementary Fig. 6b). Inhibition of cIAP1/2 by SM alone—like CDK inhibition—had a minor impact on cell death, which was

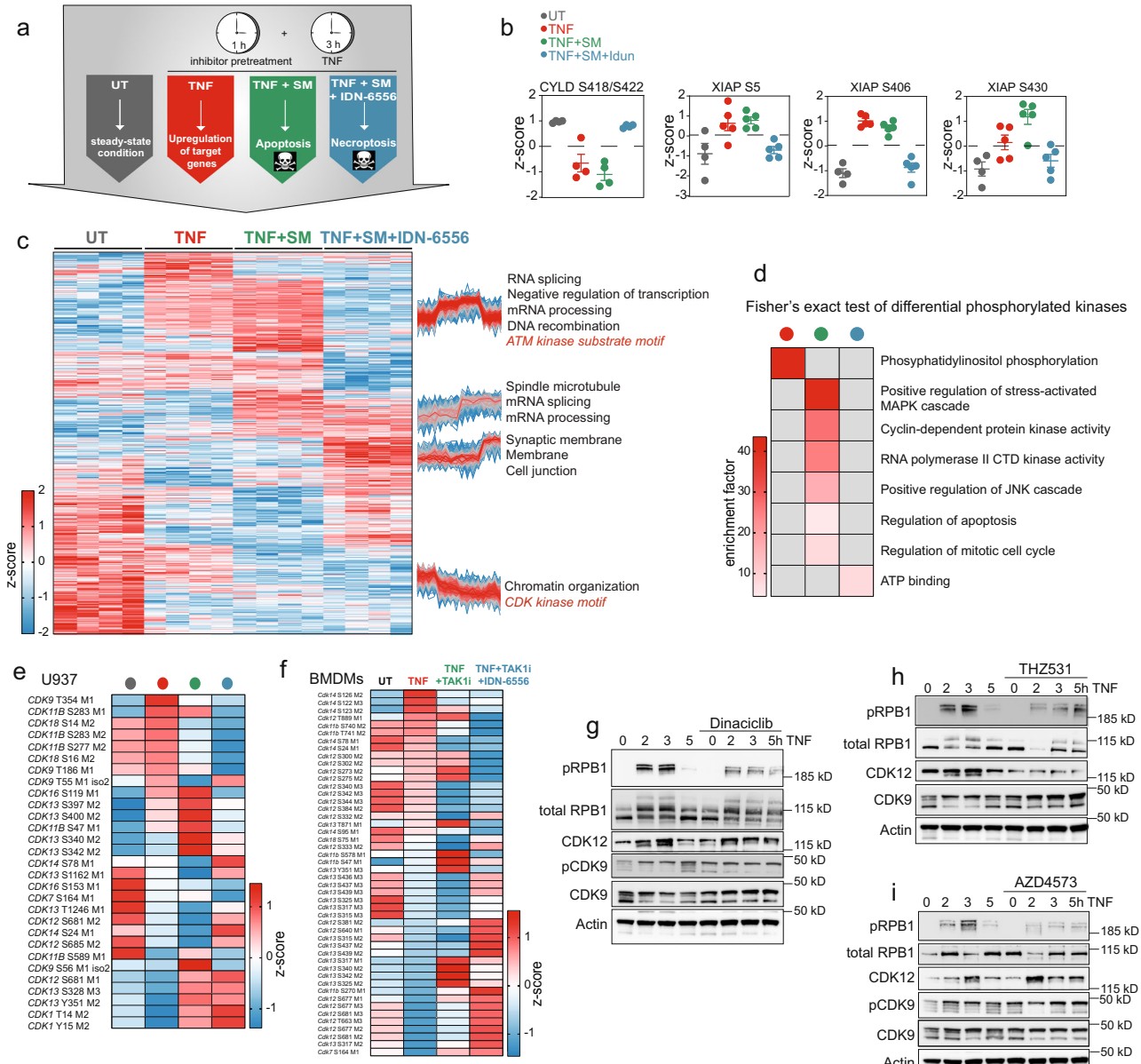

**Fig. 4 TNF-induced cell death leads to a strong RNA-processing response and regulation of the phosphorylation status of CDKs. a** Experimental scheme of U937 cells that were untreated, treated with TNF (100 ng/ml, red) alone, treated with TNF and Smac mimetic (SM/Birinapant, 1.25 μM, green) to induce apoptosis, or with TNF, SM, and IDN-6556 (Emricasan, 10 μM, blue) to induce necroptosis. **b** Z-scored phosphosite intensities of CYLD and XIAP that are regulated upon TNF treatment, TNF-induced apoptosis, and TNF-induced necroptosis (±SEM, $n = 4$ biologically independent experiments). XIAP phosphosites were retrieved from a DDA dataset (Methods). **c** Heatmap of z-scored phosphosite intensities that are one-way ANOVA significantly regulated upon TNF treatment or TNF-induced cell death (FDR < 0.05). The profiles are color-coded according to their distance from the respective cluster center (red is close to the center, blue is further away from the center). **d** Fisher's exact test on kinases with significantly regulated phosphosites upon different treatments (FDR < 0.02). **e** Heatmap of means of z-scored phosphosite intensities of CDKs that significantly changed upon treatment of U937 cells (one-way ANOVA, FDR < 0.05). **f** Heatmap of means of z-scored phosphosite intensities of CDKs that changed significantly (one-way ANOVA) upon treatment of BMDMs with TNF (red) alone, TNF, and TAK1 inhibitor (TAKi, 1 μM) to induce apoptosis (green) or TNF, TAK1i, and the caspase inhibitor IDN-6556 to induce necroptosis (blue). Untreated cells (UT) served as controls (FDR < 0.05). **g–i** Immunoblots of U937 cells stimulated with TNF as indicated or in combination with CDK inhibitors Dinaciclib (6 nM) (**g**), THZ531 (200 nM) (**h**), and AZD4573 (6 nM) (**i**) and probed with antibodies against phosphorylated (S2) and total (N-terminal) RPB1, phosphorylated (T186), and total CDK9, CDK12, and β-actin ($n = 2$ biologically independent experiments). Source data are provided as source file.

increased by their combination, and especially by the addition of TNF (Fig. 6a–d; Supplementary Fig. 6b).

To test if CDK inhibitors also enhance TNF-dependent necroptosis, we treated U937 cells with a combination of SM and the caspase inhibitor IDN-6556[52]. This showed that CDK inhibition —to a smaller degree also CDK12/13 inhibition—enhanced necroptosis in wild-type U937 cells, whereas cells lacking the

necroptotic mediators MLKL and RIPK3 failed to die (Fig. 6a–d, h; Supplementary Fig. 6c, f, g). The difference in necroptosis exacerbation between the different CDK inhibitors suggests a different mode of action of CDK12/13 inhibitors and other inhibitors for transcriptional CDKs. Whether this is based on their differential impact on FLIP isoform expression, remains to be determined. Synergistic induction of apoptosis is not restricted to

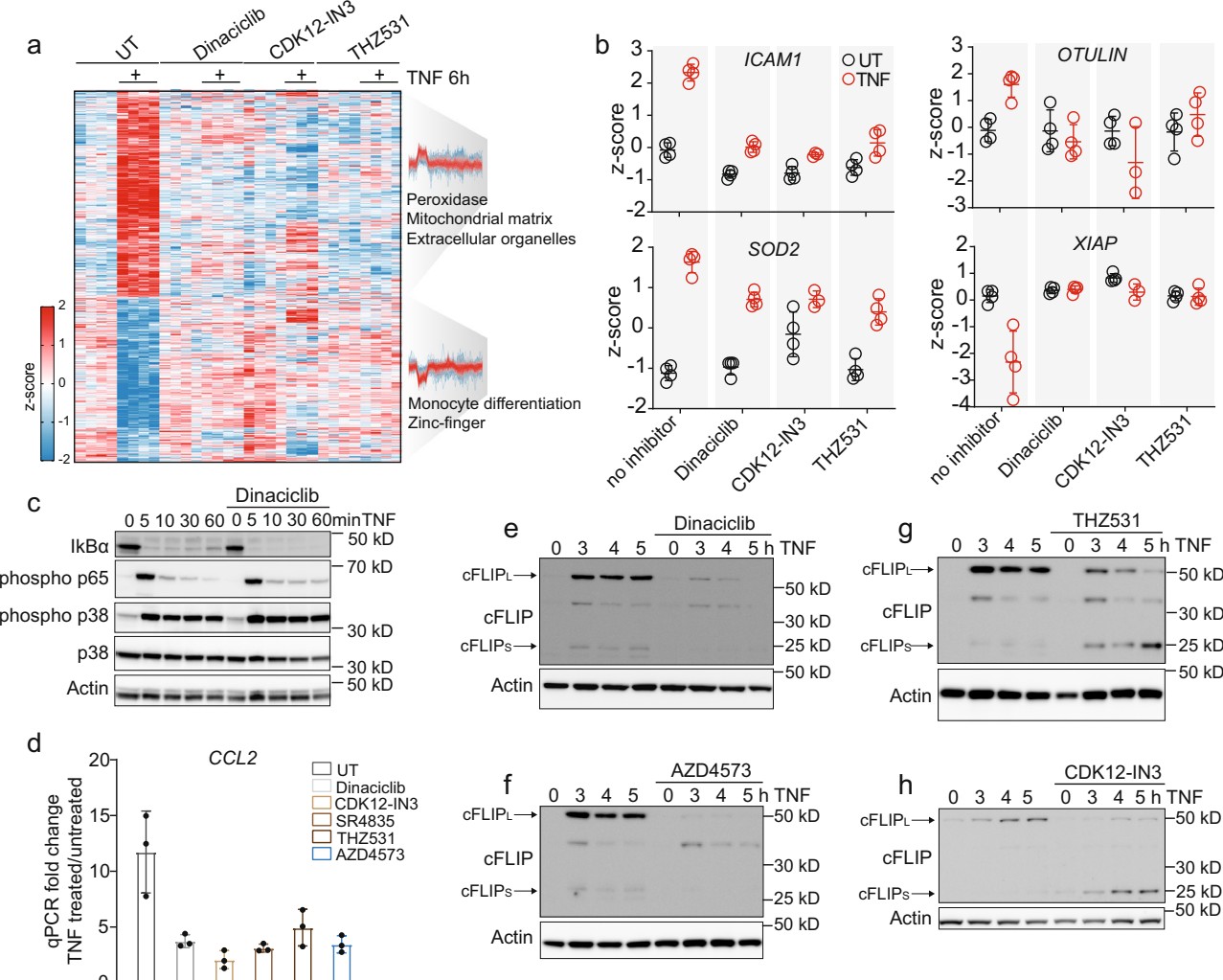

**Fig. 5 CDK9 and CDK12/13 inhibitors potently reduce transcription of TNF-target genes. a** Heatmap of z-scored protein intensities significantly changed upon TNF treatment for 6 h in U937 cells (FDR < 0.05). Cells were additionally treated with Dinaciclib (6 nM), CDK12-IN3 (60 nM), and THZ531 (400 nM). Fisher's exact test on up- and downregulated protein clusters (two-sided Student's t-test, FDR < 0.001). The profiles are color-coded according to their distance from the respective cluster center (red is close to the center, blue is further away from the center). **b** Z-scored protein levels of selected TNF-target genes and members of the TNF-signaling pathway (±SD, n = 4 biologically independent experiments). **c** Immunoblot of U937 cells treated with TNF alone and in combination with the pan-CDK inhibitor Dinaciclib. Proteins were blotted for IκBα, phosphorylated p65, phosphorylated, and total p38 and β-actin (loading control) (n = 2 biologically independent experiments). **d** qPCR of *CCL2* of U937 cells treated for 4 h with TNF alone or in combination with CDK inhibitors Dinaciclib (6 nM), CDK12-IN3 (60 nM), SR4835 (160 nM), THZ531 (200 nM) and AZD4573 (6 nM) (±SD, n = 3 biologically independent experiments). **e–h** U937 cells treated with TNF alone and in combination with CDK inhibitors (concentrations as in d) were blotted for cFLIP and β-actin (n = 2 biologically independent experiments). Source data are provided as source file.

U937 cells but also present in A549 cells, HT29 cells, U2OS cells, mouse dermal fibroblasts (MDFs), and murine BMDMs (Fig. 6i–l; Supplementary Fig. 6h–j). Unlike pan-CDK inhibitors, CDK12-specific inhibitors failed to induce synergistic cell death in BMDMs, most likely due to sustained levels of FLIP_L and the upregulation of FLIP_S (Supplementary Fig. 6j, k). TNF alone was not sufficient to induce synergistic cell death in these cells, suggesting that CDK inhibition can only potentiate but not trigger cell death (Fig. 6i–l; Supplementary Fig. 6i, j). We conclude that TNF-mediated regulation of transcriptional CDKs results in phosphorylation and regulation of their substrates, including the CTD of RPB1. This largest RNA polymerase-II subunit is necessary for transcription of prosurvival proteins like FLIP, and its phosphorylation is necessary to prevent apoptosis and necroptosis (Fig. 6m). Cell-death data in different cell lines suggest that the exacerbation of cell death by

CDK inhibitors depends on their sensitivity to TNF-mediated apoptosis and necroptosis.

## Discussion
TNF primarily exerts its functions through the transcriptional regulation of multiple target genes, which is strongly dependent on the integration of a number of different signaling events. Furthermore, the exact timing of signaling events is essential for appropriate signaling. Here, we aimed to elucidate the kinetics of TNF-induced phosphorylation events on a global scale. Our in-depth, quantitative phosphoproteome analysis revealed that early and late TNF signaling is characterized by distinct phosphorylation events, which broadly represented upstream and downstream signaling events.

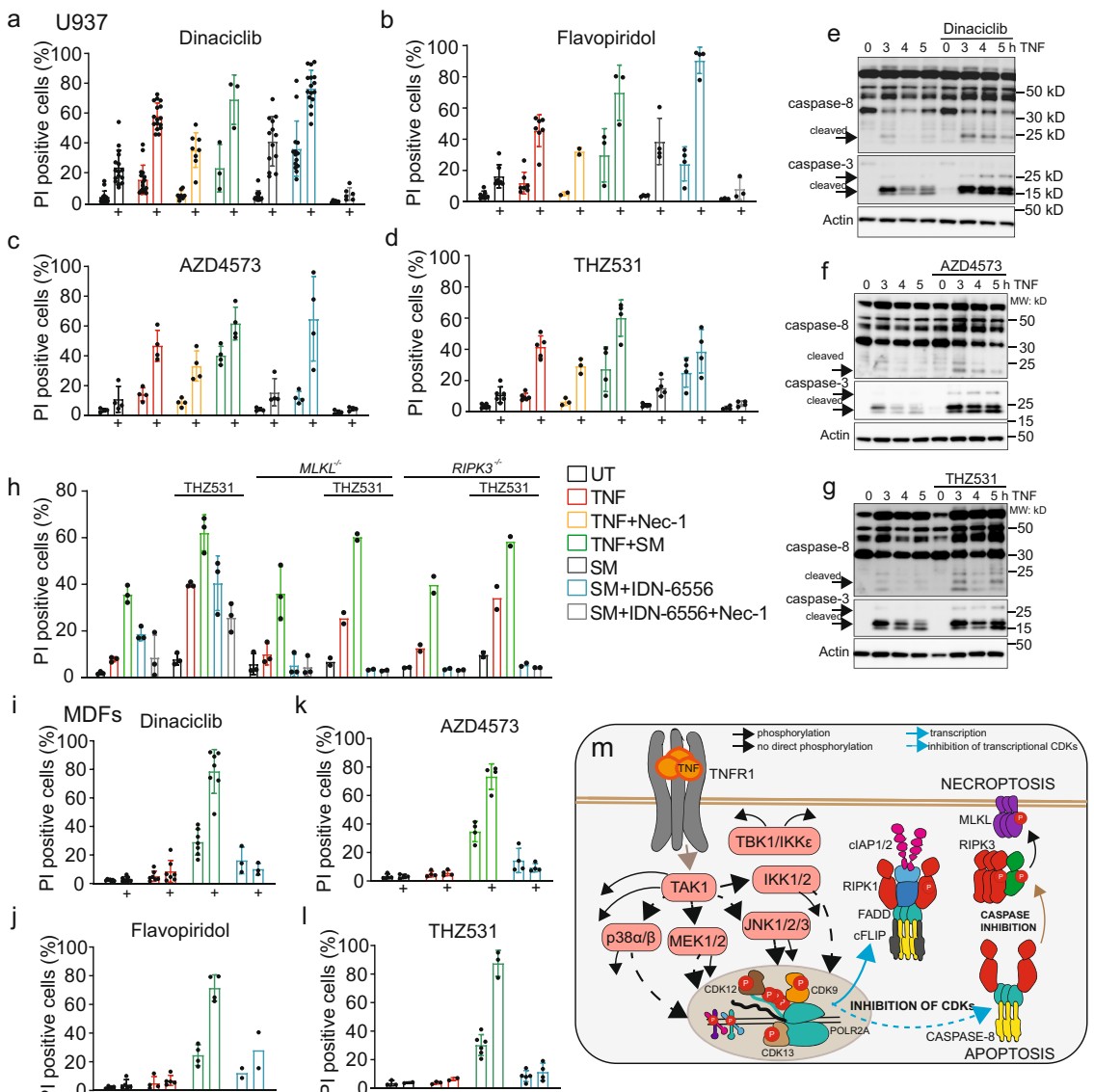

**Fig. 6 CDK9 and CDK12/13 inhibit TNF-induced cell death. a–d** Cell-death analysis by flow cytometry of propidium iodide-stained U937 cells. Cells were pretreated for one h with CDK inhibitors Dinaciclib (6 nM) (**a**), Flavopiridol (60 nM) (**b**), AZD4573 (6 nM), and (**c**) THZ531 (200 nM) (**d**) or left untreated (black) before treatment with TNF (red), TNF and necrostatin-1 (50 μM, yellow), TNF and SM (Birinapant, 1.25 μM, green), SM alone (gray), SM and IDN-6556 (10 μM, blue), or SM, IDN6556, and necrostatin-1 (light gray) for 24 h (±SD, n = 3 biologically independent experiments). **e–g** Immunoblots of U937 cells stimulated with TNF or TNF in combination with CDK inhibitors stained for caspase-3 and -8 and β-actin (n = 2 biologically independent experiments). **h** Cell-death analysis by flow cytometry of propidium iodide-stained wt and MLKL- and RIPK3-deficient U937 cells stimulated as described for 24 h (±SD, n = 2 biologically independent experiments). **i–l** Flow cytometry analysis of propidium iodide-stained wild-type MDFs treated as indicated (SM, compound A (2 μM); pan-CDK inhibitor, Dinaciclib (12 nM); CDK9 inhibitors, Flavopiridol (60 nM), AZD4573 (6 nM); CDK12/13 inhibitors THZ531 (800 nM)) (±SD, n = 2 biologically independent samples and experiments). **m** TNF-signaling scheme highlighting the role of the kinases and specifically CDKs in cell death as inferred from our data. Source data are provided as source file.

Importantly, TNF stimulation never occurs in isolation, and upon infections, TLRs and inflammasomes are activated, leading to the production of a range of cytokines. Indeed, our data showed that TNF also induced a robust and dynamic regulation of phosphorylation on proteins involved in other immune-signaling pathways. We observed TNF-induced phosphorylation events of the inflammasome components like NLRC4, adaptor proteins like MAVS, kinases like IRAK4, and phosphatases like INPP5D, and speculate that they could act to prime and interlink immune-signaling pathways to either enhance or dampen signaling strength. Our data suggest that this cross talk is most likely mediated through the activation of mutual kinases, congruent with the fact that TAK1, IKKs, p38, MEK1,2, and JNK, for

example, are activated by various immune ligands. Inhibition of the TAK1 showed that the vast majority, but not all TNF-regulated phosphorylation events are dependent on this master kinase. The few TAK1-independent phosphosites were instead regulated by IKK2, TBK1 or IKKε. It has recently been shown that TBK1 and IKKε are recruited to the TNF-signaling complex[32] where IKKε phosphorylates CYLD. Here we have identified several other likely substrates of TBK1 and IKKε, including members of the LUBAC and speculate that they localize to LUBAC's vicinity.

To identify kinases apart from IKKs and MAPKs that act downstream of TAK1 and contribute to TNF-mediated cytokine production and cell death, we analyzed the activation of kinases

upon TNF-induced apoptosis and necroptosis. This uncovered an extensive phosphoregulation of transcriptional CDKs, including CDK9, 12, and 13. Unlike the inhibition of TAK1 or IKK, CDK inhibition alone was for several cell lines not sufficient to trigger cell death upon TNF treatment. CDK9 and CDK12/13 inhibition could have a weaker impact on TNF-induced transcription compared with TAK1 and IKK inhibition or additional functions of these latter kinases on the TNF-signaling complex may directly contribute to cell-death induction independent of transcription[16].

CDK9 and CDK12/13 inhibitors are currently tested for their efficiency to kill cancer cells[53]. In particular, CDK12, which also plays a role in RNA processing/splicing and upregulation of DNA repair genes[54], has recently been reported to frequently be mutated in cancer, making it a promising cancer-therapy target[55]. Here we show that their activity strongly modulates the upregulation of most TNF-induced genes, including the caspase-8 inhibitor protein cFLIP. Reduction in CDK activity or levels induced synergistic cell death with TNF in transformed cells. Tumor therapy might profit from our finding that not only CDK9[51] but also CDK12/13 inhibitor-mediated killing of cancer cells can be enhanced by cell-death-inducing cytokines and ligands. Furthermore, CDK inhibition could also be useful in antiviral therapy. Viruses such as HIV exploit CDKs for viral transcription[56]. Therefore, targeting CDKs may both inhibit the transcription of viral genes and also enhance cell death of infected cells.

Cytokine expression induced by TNF was also heavily affected by CDK9 and CDK12/13 inhibitors. Previous studies already proposed targeting CDK9 as an anti-inflammatory treatment[57]. Here we demonstrated that CDK12/13 inhibition also potently impacts cytokine upregulation by TNF. Although transcriptional CDK inhibitors reduced the expression of proinflammatory cytokines, they increased TNF-induced cell death substantially, which in turn exacerbates inflammation[58]. This might explain increased levels of IL-6 in the pan-CDK inhibitor Flavopiridol-treated patients suffering from proinflammatory syndrome[59]. In contrast to the CDK9 and pan-CDK inhibitors, CDK12 inhibition upregulated $FLIP_S$ levels. Our observation that CDK12 inhibition failed to induce synergistic cell death in BMDMs suggests this protein as a potentially more suitable target for anti-inflammatory therapy.

In conclusion, our in-depth phosphoproteome analysis of TNF-stimulated myeloid cells elucidated the kinetics, kinase-substrate relations, and localizations of thousands of phosphorylation events. In a cell-death setting, we detected a strong regulation of transcriptional CDK phosphorylation, which we subsequently identified to be necessary for intact RPB1 phosphorylation and transcription of TNF-target genes. We propose that depending on the context, inhibitors against different transcriptional CDKs could be used as anti-inflammatory agents or tumor-cell killers. Our work provides a valuable resource of TNF-regulated phosphorylation events to the community. As importantly, it discovers an aspect of TNF signaling that is essential for intact transcription of TNF-target genes, thereby contributing to the decision between cell death and survival, autoinflammation, or an efficient response to infections.

## Methods

**Experimental model and subject details.** U937 cells were purchased from ATCC and cultured in RPMI media with 10% FCS and Pen/Strep at 37 °C and 5% $CO_2$. Mouse dermal fibroblasts (MDFs) were obtained from the epidermis and dermis of mouse tails through dispase II and collagenase digestion. Mouse dermal fibroblasts were then immortalized by infecting cells with the lentivirus SV40 large T antigen. MDFs were cultured in DME media supplemented with 10% FCS and Pen/Strep at 37 °C and 5% $CO_2$. Bone marrow-derived macrophages (BMDMs) were generated by flushing the bone marrow of murine hind bones with PBS. Cells were pelleted and were left for differentiation to macrophages in DME media with 20% L929 supernatant, nonessential amino acids, 10% FCS, and Pen/Strep at 37 °C and 5% $CO_2$ for seven days. BMDMs were kept overnight in DME media with 10% FCS and Pen/Strep at 37 °C and 5% $CO_2$ before stimulation. The animals used are bred

for scientific purposes. The research in this project does not involve experiments on an animal, but only organs harvested from dead mice. This does not require ethical approval. All animals are sacrificed prior to the removal of organs in accordance with the European Commission Recommendations for the euthanasia of experimental animals (Part 1 and Part 2). Breeding and housing as well as the euthanasia of the animals are fully compliant with all German (e.g., German Animal Welfare Act) and EU- (e.g., Directive 2010/63/EU) applicable laws and regulations concerning care and use of laboratory animals.

**Generation of cell lines.** A list of all oligonucleotides used in this study is provided in Supplementary Table 1. The XIAP knockouts were generated using two CRISPR–Cas9 RNPs (gRNA1: TTCTCTTTTTAGAAAAGG; gRNA2: GACTTTT AACAGTTTTGA). CRISPR–Cas9 RNPs were assembled by annealing synthetic, chemically stabilized crRNA:tracrRNA pairs (IDT) at 95 °C for 5 min followed by incubation at room temperature for 30 min. gRNAs were then mixed with recombinant NLS–Cas9 protein for 20 min at room temperature. Cas9 (40 pmol) was added for each 100 pmol of XIAP gRNA. RNPs were then mixed with the cell suspension and nucleofection was conducted using the SG Cell Line 4D-Nucleofector X Kit S (Lonza) on the X unit of a 4D-nucleofector (Lonza) (program FF-100). After nucleofection, cells were collected from the nucleofection cuvettes and transferred into a 96-well plate. After 48 h, U937 cells were then subject to single-cell dilution cloning in a round-bottom 96-well plate. When colonies became visible after approximately three weeks, clones were collected and subjected to MiSeq analysis as previously described[60]. In short, a genomic amplicon of 220- to 270-nt length was chosen with the nuclease-target sites located in the middle. Primers of 23 nt each with a GC content of ~50% within the entire primer sequence and a GC content of exactly 50% in the two 3′-terminal primer positions were chosen. The primer sequences were elongated by the following sequences: Fwd: ACACTCTTTCCCTACACGACGctcttccgatct—$N_{23}$ and Rev: TGACTG-GAGTTCAGACGTGTGctcttccgatct—$N_{23}$. The amplicons from each clone were then further amplified by unique barcode primers of varying sequence, pooled, and separated by gel electrophoresis. The DNA band was excised and DNA extracted by DNA gel extraction (manufacturer). Purified DNA was then sequenced by illumina MiSeq.

The CDK12 knockdown of U937 cells was achieved by lentiviral infection of three plasmids encoding for short-hairpin RNAs (Sigma Mission), which target the CDK12 mRNA for degradation (shRNA1: CCGGGCTCGGCTCTATAACTCTGAACTCGAGTTCAGAGTTATAGAGCCG AGCTTTTT, shRNA2: CCGGGCACTGAAAGAGGAGATTGTTCTCGAGAACAATCTCCTCTTTCA GTGCTTTTT, shRNA3: CCGGGATCGATGAAGGACCGGATATCTCGAGATATCCGGTCCTTCATCG ATCTTTTTG).

**Cell stimulation.** U937 cells, BMDMs, and MDFs were either left untreated or treated with TNF (100 ng/ml) alone or in combination of and IAP inhibitors/Smac mimetics (SM, Birinapant, 1.25 μM for U937 cells 5 μM for A549, HT29, U2OS and compound A, 2 μM for MDFs and BMDMs, 1 μM for A549, HT29 and U2OS cells) to induce apoptosis or with TNF, SM, and the pan-caspase inhibitor IDN-6556 (IDN-6556/Emricasan, 10 μM) to induce necroptosis. Cells were pre-treated with inhibitors for one hour [RIPK1 inhibitor] (necrostatin-1, 50 μM); TAK1 inhibitor (7-Oxozeaenol, 1 μM); IKK1/2 inhibitor (TPCA-1, 5 μM); p38 inhibitors (LY228820, 2 μM and SB203580, 10 μM); MEK inhibitor (PD0325901, 10 μM); JNK inhibitor (SP600125, 20 μM); TBK1/IKKε inhibitor (MRT67307, 2 μM); pan-CDK inhibitor (Dinaciclib: 6 nM for U937cells, 12 nM for A549, HT29, U2OS cells and MDFs, and 24 nM for BMDMs); CDK12/13 inhibitors (THZ531, 200 nM for U937, A549, HT29 and U2OS cells and 400 nM for the proteome experiment described in Fig. 5A, 800 nM for MDFs and BMDMs; SR4835, 60 nM for U937cells , 120 nM for A549, HT29 and U2OS cells and 160 nM for the qPCR experiment described in Fig. 5D, 160 nM for MDFs and 320 nM for BMDMs); CDK12 inhibitor (CDK12-IN3, 60 nM for U937 cells, 120 nM for MDFs and BMDMs); CDK9 inhibitors (Flavopiridol, 60 nM for U937 cells, MDFs, and BMDMs; AZD4573, 6 nM for U937 and for MDFs and 24 nM for BMDMs). Stimulation details and inhibitor concentrations are also indicated in Fig. legends.

**Phosphoenrichment protocol and proteome preparation for experiments and libraries.** To enrich for phosphorylated peptides, we applied the Easy Phos protocol developed in our lab[27,28]. In short, $10 \times 10^6$ U937 cells or one full 15-cm dish of BMDMs were stimulated, washed three times with ice-cold TBS, lysed in 2% sodium deoxycholate (SDC) and 100 mM Tris-HCl [pH 8.5], and boiled immediately. After sonication, protein amounts were adjusted to 1 mg using the BCA protein assay kit. Samples were reduced with 10 mM tris(2-carboxy(ethyl)phosphine) (TCEP), alkylated with 40 mM 2-chloroacetamide (CAA), and digested with trypsin and lysC (1:100, enzyme/protein, w/w) overnight. For proteome measurements, 20 μg of the peptide was taken and desalted using SDB-RPS-stage tips. About 500 ng of desalted peptides were resolubilized in 5 μl 2% ACN and 0.3% TFA and injected into the mass spectrometer. For phosphoenrichment, iso-propanol (final conc. 50%), trifluoroacetic acid (TFA, final conc. 6%), and

monopotassium phosphate (KH$_2$PO$_4$, final conc. 1 mM) were added to the rest of the digested lysate. Lysates were shaken, then spun down for 3 min at 2000 × g, and supernatants were incubated with TiO$_2$ beads for 5 min at 40 °C (1:10, protein/beads, w/w). Beads were washed five times with isopropanol and 5% TFA, and phosphopeptides were eluted off the beads with 40% acetonitrile (ACN) and 15% of ammonium hydroxide (25% NH$_4$OH) on C8-stage tips. After 20 min of SpeedVac at 45 °C, phosphopeptides were desalted on SDB-RPS-stage tips and resolubilized in 5 μl 2% ACN and 0.3% TFA and injected in the mass spectrometer.

To generate the library for phosphoproteome data independent acquisition (DIA) measurements, U937 cells were treated with TNF (100 ng/ml) in the presence of phosphatase inhibitors Sodium orthovanadate (1 mM) and Calyculin A (50 ng/ml) for 15 min. Cells were washed with ice-cold TBS and lysed in 2% SDC with 100 mM Tris-HCl [pH 8.5]. After boiling, sonication, reduction, alkylation, and overnight digestion (as described above), lysates were desalted on a Sepax Extraction column (Generik DBX), and 3 mg of desalted peptides were fractionated into 84 fractions on a C18 reversed-phase column (4.6 × 150 mm, 3.5 -μm bead size) under basic conditions using a Shimadzu UFLX operating at 1 ml/minute. Peptides were separated on a linear gradient consisting of buffer A (2.5 mM ammonium bicarbonate in MQ) and 2.5–44% buffer B (2.5 mM ABC in 80% ACN) for 64 min and 44–75% buffer B for 5 min before a rapid increase to 100%, which was kept for 5 min. Fractions were subsequently concatenated into 36 fractions and lyophilized. Phosphopeptides of these fractions were enriched using the phosphoenrichment protocol described above and run in data-dependent acquisition (DDA) with the same gradient (70 min), the samples of the DIA experiments were run.

To generate the library for the proteome DIA measurements, U937 cells were lysed in 2% SDC with 100 mM Tris-HCl [pH 8.5], boiled, sonicated, reduced, alkylated, and digested overnight before desalting using SDB-RPS cartridges. About 100 μg of peptides were fractionated into eight fractions by high-pH reversed-phase chromatography. Fractions were concatenated automatically by shifting the collection tube every 120 s and subsequently dried in a SpeedVac. Peptides of these fractions were run in DDA mode with the same gradient (120 min) as the samples of the proteome DIA experiments.

**Crude fractionation.** After stimulation, U937 cells (100 × 10$^6$ cells) were washed with TBS and lysed in 4 ml of lysis buffer (250 mM sucrose, 25 mM Tris-HCl [pH 7.5], 5 mM KCl, 3 mM MgCl$_2$), 0.2 mM EDTA with protease- and phosphatase inhibitors (sodium pyrophosphate (12.5 mM = 50 × stock), β-glycerophosphate (2.2 g/10 ml = 1000 × stock), and sodium orthovanadate (1.84 g/100 ml = 100 × stock)). Cells were broken up in the cell homogenizer (isobiotec) with 12 strokes. To 20% of lysate, SDS was added to a final concentration of 1% SDS and proteins were precipitated with acetone (final conc. 80%) for total phosphopeptide analysis. The rest of the lysate was spun down at 900 × g for 10 min. The pellet represents the nucleus fraction and was resuspended in 1% SDS before acetone precipitation. The supernatants were spun down for 40 min at 78,400 × g using a Beckmann Coulter ultracentrifuge. The pellet, which contains the membrane fraction, was resuspended in 400 μl of 1% SDS before the addition of acetone, while 1 ml of 2% SDS was added to the supernatant containing the cytosol fraction before protein precipitation with acetone. Proteins were precipitated overnight, and the next day washed with 80% acetone before resuspension in 4% SDC. The phosphoenrichment (1 mg/sample) and protein preparation (30 μg/sample) of all fractions and conditions were performed as described above.

**Western blotting.** One million U937 cells were stimulated, washed in PBS, and lysed in buffer (1% IGEPAL, 10% glycerol, 2 mM EDTA, 50 mM Tris, pH 7.5, and 150 mM NaCl) supplemented with protease inhibitors (Sigma-Aldrich, 4693159001). Lysates were kept on ice for 20 min and centrifuged at 16,100 × g for 15 min. After centrifugation 2 × SDS sample-loading buffer (450 mM Tris-HCl, pH 8, 60% (v/v) glycerol, 12% (w/v) SDS, 0.02% (w/v) bromophenol blue, and 600 mM DTT) was added to the supernatant before boiling of the lysate. Proteins were separated on 12% Novex Tris-glycine gels (Thermo Fisher Scientific, XP00120BOX) and transferred onto PVDF membranes (Merck Millipore, IPVH00010) or Nitrocellulose membranes (Amersham, 10600002). Membranes were blocked in 5% BSA in PBST, and antibodies were diluted in 2% BSA in PBST. Antibodies used for immunoblotting were as follows (diluted 1:1000): anti-human caspase-8 (MBL, M058-3), anti-cleaved human caspase-3 (Cell Signaling Technology CST, 9661), anti-human MLKL (Merck Millipore, MABC604), phospho anti-human RPB1 S2 (Millipore, 04-1571), anti-human RPB1 (CST, D8L4Y), phospho anti-human p65 (CST, 3033 P), anti-human IκBα (CST, 9242), phospho anti-human p38 (CST, 9215), anti-human p38 (CST, 9212), anti-human CDK12 (CST, 11973), phospho anti-human CDK9 (CST, 2549), anti-human CDK9 (CST, 2316), anti-human A20 (Santa Cruz, sc-166692), anti-human MCL1 (CST,4572), anti-human XIAP (MBL, M044-3), anti-human FLIP (CST, 56343) and anti-human β-actin (CST, 4967).

**Elisa.** About 1 × 10$^4$ BMDMs were plated in 96 wells stimulated with CDK inhibitors for one hour before the addition of TNF (100 ng/ml). After 24 h, the supernatants were spun down, and ELISA was performed according to the supplier's protocol.

**Cell-death analysis.** About 7 × 10$^4$ U937 cells were plated in 96 wells, and BMDMs, A549, HT29, U2OS cells and MDFs were plated in 24 well plates and treated with TNF (100 ng/ml), IDN-6556 (μ10 μM), necrostatin-1 (Nec-1, 50 μM), Birinapant (SM, 1.25 μM for U937, 5 μM for A549, HT29 and U2OS cells), and compound A (SM, 2 μM for MDFs and BMDMs, 1 μM for A549, HT29 and U2OS cells). Cell death was measured by propidium iodide incorporation using flow cytometry (FACS Attune NxT) and analyzed using FlowJo (version 10.6.1) and Graphpad Prism (version 8.4.3).

**qPCR.** RNA was isolated from 2 × 10$^6$ cells with the RNeasy Plus Mini kit (Qiagen) and reversely transcribed with SuperScript III (Invitrogen). cDNA was amplified with SYBR Green on a Biorad C1000 Thermal Cycler. Primers used were for *CCL2* (forward: CCTAGGAATCTGCCTGATAATCGA, reverse: TGGGGATATACC ATGCATACTGAGATG), for *CXCL10* (forward: GAAATTATTCCTGCAAGCC AATTT, reverse: TCACCCTTCTTTTTCATTGTAGCA) and for *GAPDH* (forward: GTCTCCTCTGACTTCAACAGCG, reverse: ACCACCCTGTTGCTGT AGCCAA). Fold induction compared with untreated controls was calculated by the delta-delta CT method.

**Chromatography and mass spectrometry.** Samples were loaded onto 50-cm columns packed in-house with C18 1.9 μM ReproSil particles (Dr. Maisch GmbH), with an EASY-nLC 1000 system (Thermo Fisher Scientific) coupled to the MS (Q Exactive HFX, Thermo Fisher Scientific). A homemade column oven maintained the column temperature at 60 °C. Peptides were introduced onto the column with buffer A (0.1% formic acid), and phosphopeptides for data-independent acquisition were eluted with a 70-min gradient starting at 3% buffer B (80% ACN, 0.1% formic acid) and followed by a stepwise increase to 19% in 40 min, 41% in 20 min, 90% in 5 min, and 95% in 5 min, at a flow rate of 300 nL/min, while peptides for proteome analysis were eluted with a 120-min gradient starting at 5% buffer B (80% ACN, 0.1% formic acid) followed by a stepwise increase to 30% in 95 min, 60% in 5 min, 95% in 2 × 5 min and 5% in 2 × 5 min at a flow rate of 300 nL/min. Phosphopeptides for data-dependent acquisition were eluted with a 140-min gradient starting at 5% buffer B (80% ACN, 0.1% formic acid) followed by a stepwise increase to 20% in 85 min, 40% in 35 min, 65% in 10 min, and 80% in 2 × 5 min at a flow rate of 300 nL/min.

A data-independent acquisition MS method was used for proteome and phosphoproteome analysis in which one full scan (300–1650 *m/z*, R = 60,000 at 200 *m/z*) at a target of 3 × 10$^6$ ions was first performed, followed by 32 windows with a resolution of 30,000 where precursor ions were fragmented with higher-energy collisional dissociation (stepped collision energy 25%, 27.5%, and 30%) and analyzed with an AGC target of 3 × 10$^6$ ions and a maximum injection time at 54 ms in profile mode using positive polarity. Samples for libraries were run in data-dependent acquisition mode with the same gradients.

For the proteome/phosphoproteome libraries, samples were measured in data-dependent acquisition with a (TopN) MS method in which one full scan (300–1650/1600 *m/z*, R = 60,000 at 200 *m/z*) at a target of 3 × 10$^6$ ions was first performed, followed by 15/10 data-dependent MS/MS scans with higher-energy collisional dissociation (target 10$^5$ ions, maximum injection time at 28/60 ms, isolation window 1.4/1.6 *m/z*, normalized collision energy 27%, and R = 15,000 at 200 *m/z*). Dynamic exclusion of 30 s was enabled.

For phosphoproteome measurements in data-dependent acquisition, a (TopN) MS method was used in which one full scan (300–1650 *m/z*, R = 60,000 at 200 *m/z*, maximum injection time 120 ms) at a target of 3 × 10$^6$ ions was first performed, followed by 10 data-dependent MS/MS scans with higher-energy collisional dissociation (AGC target 10$^5$ ions, maximum injection time at 120 ms, isolation window 1.6 *m/z*, normalized collision energy 27%, and R = 15,000 at 200 *m/z*). Dynamic exclusion of 40 s and the Apex trigger from 4 to 7 s was enabled.

**Quantification and statistical analysis.** For the experiment measured in DDA mode, MS raw files were processed by the MaxQuant version 1.5.0.38[61] and fragments lists were searched against the human UniProt FASTA database (21,039 entries, August 2015)[62] with cysteine carbamidomethylation as a fixed modification and N-terminal acetylation and methionine oxidations as variable modifications. For phosphoproteome analysis, we also added serine/threonine/tyrosine phosphorylation as variable modification. We set the false-discovery rate (FDR) to less than 1% at the peptide and protein levels and specified a minimum length of seven amino acids for peptides. Enzyme specificity was set as C-terminal to arginine and lysine as expected using trypsin and lysC as proteases and a maximum of two missed cleavages.

For experiments measured in DIA mode, MS raw files were processed by the Spectronaut software version 13 (Biognosys)[63]. First, hybrid libraries were generated in Spectronaut Pulsar by combining the DDA runs of fractionated samples of either proteome or phosphoproteome with the DIA runs of the respective experiments. Human (21,039 entries, additional 74,013 entries, 2015) and mouse uniport FASTA databases (22,220 entries, 39,693 entries, 2015) as forward databases were used. To generate phosphoproteome libraries, serine/threonine/tyrosine phosphorylation was added as variable modification to the default settings, which include cysteine carbamidomethylation as fixed modification and N-terminal acetylation and methionine oxidations as variable

modifications. The maximum number of fragment ions per peptide was increased from 3 to 15. The false discovery rate (FDR) was set to less than 1% at the peptide and protein levels and a minimum length of seven amino acids for peptides was specified. Enzyme specificity was set as C-terminal to arginine and lysine as expected using trypsin and LysC as proteases and a maximum of two missed cleavages. To generate proteome libraries, default settings were used. The experimental DIA runs were then analyzed against the hybrid library by using default settings for the analysis of the proteome, and for the analysis of the phosphoproteome samples the localization cutoff was set to 0. BMDM raw files were analyzed via directDIA. For phosphoproteome analysis, serine/threonine/tyrosine phosphorylation was additionally set as variable modification and localization cutoff was set to 0.

All bioinformatics analyses were done with the Perseus software (version 1.6.2.2)[64]. For phosphosite analysis spectronaut normal report output tables were collapsed to phosphosites and the localization cutoff was set to 0.75 using the peptide collapse plug-in tool for Perseus[65]. It collapses phosphoions to phosphosites. Importantly, it does not sum up the intensities of a phosphosite on peptides, if different phosphorylations are also present. For example, the intensity of MAPK14_T180_M1 and MAPK14_T180_M2 (for MAPK14_T180_Y182) differs. For each phosphosite on a multiple phosphorylated peptide, we receive a row with the same intensities as these phosphorylations are localized on the same peptide. While MAPK14_T180_M1 represents the singly phosphorylated peptide, MAPK14_T180_M2 and MAPK14_Y182_M2 share the same intensity as they represent the two phosphosites on the same peptide. Different phosphosites on the same peptide can have slightly different fold changes due to imputation. Also, each collapse key (gene_position_multiplicity) is unique, which means that if a phosphosite is present on two peptides that carry a different phosphorylation, just one row will be assigned. Phosphosites located on phosphopeptides with more than two phosphorylations are labeled with a multiplicity of 3 (M3). Summed intensities were log2-transformed. Samples that did not meet the measurement quality of the overall experiment were excluded. For the 8-min time point of the time-course experiment, we started with fewer replicates. Quantified proteins were filtered for at least 75% of valid values among three or four biological replicates in at least one condition. Missing values were imputed and significantly up- or downregulated proteins were determined by multiple-sample test (one-way analysis of variance (ANOVA), FDR = 0.05) and Student's t-test (two-sided) (FDR = 0.05). For Fig. 2e, we used two imputations to additionally obtain low abundant TNF-induced phosphosites (normal imputation: width 0.3, downshift 1.8, low imputation: width 0.15, downshift 3).

n represents replicates of the same cell line stimulated separately. Further statistical details of experiments can be found in the figure legends.

The 1D annotation-enrichment analysis detects whether expression values of proteins belonging to an enrichment term (here we used keywords GOCC, GOMF, GOBP, and KEGG name) show a systematic enrichment or deenrichment compared with the distribution of all expression values[66].

Fisher's exact tests were performed to detect the systematic enrichment or deenrichment of annotations and pathways by analyzing proteins whose levels or phosphorylation levels are significantly regulated upon different conditions (we used keywords GOCC, GOMF, GOBP, and KEGG name). The Benjamini–Hochberg FDR represents the degree of significance and the enrichment factor the level of enrichment compared with the background.

Kinase-motif enrichment analysis (Supplementary Fig. 1f) was performed by loading significantly (FDR < 0.05) up- and downregulated phosphosites onto the website: http://phosfate.com/profiler.html[67].

Networks (Supplementary Fig. 1e, Fig. 2d) were generated using STRING[68] by uploading significantly changing phosphosites, setting a medium-confidence score (0.4), and including the active-interaction sources: Textmining, Experiments, and Databases.

**Dashboard for data visualization and manual inspection**. For the TNF time-course analysis and visualization, the median z-score and standard deviation for each phosphosite and time point were calculated per condition. The coregulation analysis was performed by calculating all pairwise Pearson correlations between phosphopeptides across the sampled time points. The maximum q-value filter in the different analyses allows to filter for peptides or proteins that were significant in at least one ANOVA-based comparison across conditions. A q-value of 0.01 means that maximally 1% of all results that are reported to be statistically significant are estimated to be false positives.

For sequence visualization and protein-domain annotation, each phosphosite location was mapped to its respective protein sequence stored in the fasta file that was used for MS/MS data analysis (human fasta, downloaded 2015). The protein sequences for visualization were obtained using the 'fasta' functions from pyteomics[69,70]. Information about protein domains was retrieved from UniProt (https://www.uniprot.org/, accessed 22.06.2020 for human and 12.07.2020 for mouse), including the following categories: 'Chain', 'Domain', 'Alternative sequence', 'Propeptide', 'Signal peptide', and 'Transit peptide'.

Data preprocessing and visualization for the dashboard was performed using the python programming language (v3.7.7). The following libraries were utilized for data processing: numpy, pandas, scipy, re, and pyteomics[69,70]. Several libraries from the HoloViz family of tools were used for data visualization and

creation of the dashboard, including panel, holoviews, param, bokeh, plotly, and matplotlib.

**Reporting summary**. Further information on research design is available in the Nature Research Reporting Summary linked to this article.

## Data availability
The MS-based proteomics data have been deposited to the ProteomeXchange Consortium via the PRIDE partner repository and are available via ProteomeXchange with identifier PXD021366[71]. Source data are provided with this paper. The website for the investigation of TNF-regulated phosphopeptides is available on http://tnfviewer.biochem.mpg.de/. Information about protein domains was retrieved from UniProt (https://www.uniprot.org/, accessed 22.06.2020 for human and 12.07.2020 for mouse). Source data are provided with this paper.

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

## Acknowledgements

We thank Bianca Splettstoesser for extensive technical support and Igor Paron, Christian Deiml, Johannes B. Müller, Antonio Piras, and Gabriele Sowa for technical assistance, Meera Phulphagar, Julia Schessner, and Georg Borner for their scientific input. Margaret Jackson helped with experiments. Ashok Jayavelu provided scientific input and reagents. John Silke and Najoua Lalaoui (Walter and Eliza Hall Institute) provided Birinapant and compound A. Martin Spitaler and Markus Oster (imaging facility, Max Planck Institute of Biochemistry) helped with flow cytometry. Medini Steger provided helpful comments to the paper. We thank all members of the Department of Proteomics and Signal Transduction at the Max Planck Institute of Biochemistry in Martinsried for their help and discussions. This work was supported by the Max Planck Society for the Advancement of Science and a Marie Sklodowska-Curie Actions Individual Fellowship awarded to M.C.T (746329). This project has also received funding from the European Union's Framework Programme for Research and Innovation Horizon 2020 (2014–2020) under the Marie Skłodowska-Curie Grant Agreement No. 754388 and from LMU Munich's Institutional Strategy LMUexcellent within the framework of the German Excellence Initiative (No. ZUK22). I.B. was supported by a Swiss National Science Foundation Postdoc.Mobility fellowship (P400PB_191046). V.H. is supported by the German Research Foundation (Deutsche Forschungsgemeinschaft) CRC 1403—project number 414786233.

## Author contributions

M.C.T. and M.M. conceived the project and wrote the paper. M.C.T. designed, performed all experiments, and analyzed most data (except ELISA). I.B. analyzed data and generated the website. C.A.S. performed the ELISA experiment, generated XIAP knock out cells, and edited the paper. V.H. provided reagents and edited the paper.

## Funding

## Competing interests

The authors declare no competing interests.
