## [Peer Review File · Nature Communications]

REVIEWER COMMENTS

Reviewer #1 (Remarks to the Author):

Tanzer and colleagues provide a rich proteomic resource for scientists interested in TNF signaling in myeloid cells. In the latter part of the paper, they focus on transcriptional CDKs, which were activated by TNF and mediated phosphorylation of a subunit of RNA Pol II. Activation of pro-survival genes in this manner is proposed to limit TNF-induced cell death. Experiments are well controlled and I support publication. I have only a few suggestions for the authors to consider below.

Specific comments:

1) The authors highlight signaling downstream of TNFR1, but can they link any of the phosphorylation events to NIK activation downstream of TNFR2?

2) Fig 2a suggests that activation of TBK1/IKKe in response to TNF is divorced from activation of TAK1 and IKK1/2. This differs from the model of Philip Cohen's group in their 2011 paper (PMID: 21138416), which states in the abstract "the TNF α -induced activation of the IKK-related kinases is mediated solely by the canonical IKKs." Perhaps their results differ because they used MEFs rather than myeloid cells? Can the authors comment on whether they think their results in myeloid cells are more broadly applicable to other types of cells.

3) Fig 6 and EDF. 6 focus on regulation of cFLIP as a potential mechanism by which the CDKs suppress cell death. What about the regulation of other highly induced genes such as A20/TNFAIP3, which suppresses the transition from TNFR1 complex 1 to death-inducing complex II?

Reviewer #2 (Remarks to the Author):

Tanzer et al. provide a thorough mass spec analysis revealing a large number phosphorylation sites in TNF-stimulated U937 cells. The results are of potentially high interest to scientists who study TNF signaling. The study also examines phosphorylation events that depend on the activity of transcriptionally acting CDKs, further broadening the relevance of some of the identified phosphorylation sites. The amount of generated data is impressive, however, the mass spec work is mostly based on the analysis of one cell line, U937 cells, in which the authors identify a number of novel phosphorylation sites (figs. 1-4). It seems that most of the phosphorylation sites detected in U937 cells cannot be detected in BMDMs or MEFs. Are the findings obtained in U937 cells which are reported in figs. 1-4 representative of TNF signaling in general or could they be specific to this cell line? By the end of figure 4 the authors have essentially provided a list of several potentially interesting phosphorylation events that can now be validated functionally in U937 cells and in other cells. However, such a validation is not presented. The only attempt at validation is presented in extended figure 4b and shows no significant effect of XIAP phosphorylation sites on TNF induced cell death. It also remains somewhat unclear how this validation attempt was conducted. How were the XIAP KO cells made? Has this been done by CRISPR/CAS9 and how were the KO cells reconstituted? What are the levels of the reconstitution that were achieved with the individual mutants and how different are these levels to reconstitution with wild type and the expression level in the parental cell line before KO? Thus the first part of the study offers many potentially meaningful phosphorylation sites but none of them is adequately validated functionally with respect to TNF signaling.

The study next embarks on a different direction and examines the regulation of TNF target gene expression mediated by transcriptional CDKs. This part is biochemically explored in more detail, yet the knowledge of transcriptional CDK inhibition and sensitization to cell death induction by TNF superfamily members is not novel and has been reported in several other studies before. The authors reference these studies appropriately but their work fails to provide an advance beyond the current understanding of this process.

The following major points need to be addressed by the authors:

1. Figs. 1-4 show results on TNF stimulation of U937 cells. Are any of the newly identified phosphorylation sites also observed in other cell types or cell lines, including in mouse cells? A biochemical and functional validation of specific phosphorylation events on a few of the newly identified targets would provide for a useful validation of the results obtained by mass spectrometry.

2. In several instances the authors use kinase inhibitors which are often only selective and not specific. Have the authors compared them to loss of expression of the main targets of such inhibitors?

3. One of the aims of the study is to evaluate how transcriptional CDKs are regulated by phosphorylation. The authors should therefore validate the reported findings using suitable systems. How does mutation of the main phosphorylation sites identified for transcriptional CDKs affect TNF signaling? Additionally, the authors should repeat at least some of the phosphoproteomic experiments in the presence of the CDK inhibitors. In figure 5 they only provide a proteomic analysis of TNF-treated U937 cells in the presence or absence of the inhibitors and the data provided in Extended Data Fig. 5a cannot be compared with the phosphoproteomic analysis carried out in figure 5a, not only because the authors only show selective changes in Extended Data Fig. 5a, but also, and most importantly, because the phosphorylation sites identified upon TNF stimulation are not consistent between the two figures. The authors should carry out a wider screen using different cell types to evaluate how reproducible the observed phosphorylation events are since the comparison between U937 and BMDM already resulted in different outcomes. As an example, TNF stimulation induced a different pattern of phosphorylation of transcriptional CDKs in BMDMs versus U937 cells (comparison between figs. 4e and 4f), and in BMDMs no phosphorylation events were detected for CDK9 upon TNF stimulation, yet there was a strong upregulation of phosphorylation of CDK14, whose role in transcription is not well characterized. The authors do not comment on this result but instead mainly focus on the effects observed in U937 cells. How the authors explain the discrepancy between these two cellular systems? Are the observed effects cell-type specific?

4. In fig. 5 the authors investigated if and how CDK inhibitors affect TNF mediated regulation of target genes at the transcriptional level. It is not surprising that Dinaciclib, CDK12-IN3 and THZ531 reduce transcription of TNF target genes, as it is known that CDK9, 12 and 13 regulate transcriptional elongation so that their inhibition would affect productive transcription, including the transcription of TNF target genes. Furthermore, the authors only show a reduction in the levels of MCP-1 mRNA in U937 cells and of IP10 release for BMDMs (Extended Data Fig. 5a) upon stimulation with TNF in the presence of the different CDK inhibitors. It is unclear why the authors chose to detect and compare two different cytokines which makes it difficult to assess the impact of CDK inhibitors on TNF-induced cytokine production, especially because from the data provided it seems that the different CDK inhibitors exert different regulatory functions in the two cellular systems employed. Additionally, fig 5e-h and fig 6 lack novelty. As correctly cited by the authors, it has already been shown that inhibition of the transcriptional CDK9 affects the expression level of the anti-apoptotic protein FLIP and that it was therefore expected that TNF-induced FLIP upregulation can be prevented by inhibition of transcriptional CDKs and that this would enhance TNF induced cell death.

5. The authors state “Our data thus suggests a potential association of these proteins with the LUBAC complex upon TNF signaling” (page 7, line 30). However, the phosphorylation of ARFGAP2, IQSEC1 and FERMT3 are suppressed by MRT67307 treatment prior to TNF stimulation. Therefore

these results do not show a direct link to LUBAC as these proteins could also interact with another component of the TNF receptor complex I or form part of downstream signaling pathways influenced by TBK1/IKKε.

6. In Extended Data Figure 4 the authors went on to identify the different phosphorylation sites on known components of the TNF signaling pathway. Here they mainly focus on the identification and validation of XIAP phosphorylation sites by knockout and reconstitution with phospho-ablative or phospho-mimetic approaches. However, it is difficult to assess the results as neither data on the efficiency of XIAP knockout in U937 nor detailed explanation of the methodology used to achieve the XIAP knockout and reconstitution are provided in the materials and methods section. The authors should assess this effect in a different cellular system perhaps in cells of epithelial origin.

7. Based on the results provided in fig. 4g-i the authors claim that TNF stimulation induces increased CDK9 phosphorylation which in turn correlates with increased phosphorylation of the CTD of RPB1 at Ser-2, and this could be prevented by the different CDK inhibitors employed. It seems surprising however that at time point "0" there was no detectable Ser-2 phosphorylation of the CTD of RPB1 whereas in the same condition the authors were able to detect the activating phosphorylation of CDK9 (pCDK9) at position T186. Moreover, inhibition of CDK9 with Dinaciclib or AZD4573 induced an increase in the levels of pCDK9 independently of TNF stimulation. How the authors explain this and what happens to pCDK9 upon treatment with the CDK12 inhibitor THZ531?

8. In fig. 6 the authors show that transcriptional CDK inhibitors synergize with TNF in U937 but do not discuss that also independent of TNF and SM the combination of SM with IDN-6556 results in a synergistic cell death induction in U937. The authors should clarify this. Importantly, in the text the authors stated that "This synergistic cell death is not restricted to U937 cells but also present in mouse dermal fibroblasts (MDFs) and murine BMDMs" (page 18, line 24). However, there are no results in the present study supporting this statement as the data provided in fig. 6a-i shows a different pattern of sensitivity for the treatment with TNF/CDK inhibitors in the cellular systems employed since mouse dermal fibroblasts do not show this strong sensitivity to TNF/CDK inhibition treatment observed in U937 cells. How the authors explain these differences? Are the synergistic effects observed only specific for U937 cells?

Minor points which should be addressed are:

- In the legend of fig. 5 the authors should mention the concentration of AZD4573 used.

- The statement "IKKε phosphorylates and activates the deubiquitinase CYLD" is not correct. The authors refer to Hutti et al. (2009) who published that IKKε can phosphorylate CYLD at Ser418 and thereby decrease its activity.

Reviewer #3 (Remarks to the Author):

In the paper by Tanzer et al. the authors applied quantitative phosphoproteomics to dissect the dynamic and subcellular phosphoproteomic alteration during the course of TNF stimulation in myeloid cells. Motif enrichment analysis was performed for these altered phosphorylation event to identify upstream kinases. Compared to previous studies on TNF signaling, this study achieved much enhanced phosphoproteomic coverage to uncover the systems map of the TNF signaling. A very interesting finding inferred from the phosphoproteomics profiling is the discovery of the role of CDK kinase activity in TNF-induced cell death. Two cell types were studied in both phosphoproteomics and cell biology experiments, yet some of the results presented seem to be different, which raised the concern whether the findings are general in TNF signaling and not cell type-dependent. Nevertheless, this study provides rich resource to show the two dimensional systems map of phosphorylation-mediated TNF signaling in time and subcellular space. Given the clinical significance of CDK family and TNF signaling in inflammation and cancer therapy, these findings will be of significant interests. However, a number of significant issues require clarification to further improve the manuscript.

1. The use of data-independent acquisition (DIA) provided good overall phosphoproteomics profiling in this work. DIA is known to be a more reliable method compared to conventional approach. How is the reproducibility in replicate runs and overlap between different time point? The Authors have to describe the quantification results of each time point.

2. While it is understood that the phosphopeptide identification may generate multiple phosphorylation sites per peptide, can the Author explained more clearly on how the 60,000 phosphopeptides map to 28,000 class 1 sites ? I have one major concern on the assignments of phosphosites with different sites in the nearby location. In all of the heatmap results (eg, Fig. 2 e-f, Extended Data Fig. 2C, Fig. 3g-l, Fig. 4), most of the proteins with list of multiple sites on the same peptides have identical intensity in the heatmap. It is less likely that the nearby sites have identical alteration. Are these from the same peptides with few phosphosites which cannot be distinguished for the exact phosphosites? Such condition has to be differentiated from the one that has unambiguously localized phosphosites on the same peptides. The identification results and figures in this manuscript have to be clarified and revised.

3. The regulation of a phosphosite may be contributed from either the protein expression or phosphorylation degree. For the quantitation result shown in this study, were they normalized to the change of protein expression ?

4. The dynamic phosphoproteomic profiles at 15 min of TNF treatment was further analyzed in the murine bone marrow derived macrophages (BMDMs), in which the phenotypic nature is different than the myeloid cells. The immune response-related functional terms enriched from both cell systems are different. In the validation results of TNF-induced cell death, the Authors only mentioned that the experiment was also performed in BMDM cells. No further elaboration was made to link the result in the selected examples in the U937 cells. Whether the result from BMDM cells provide additional insight is not clear. The Authors have to clearly justify the rationale of using these two cell types. To dissect the common and unique TNF-induced cascades, comparison on the temporal response of both cell types at 15 minutes and examined whether these altered expression can be restored at later time point will be critical information.

5. In this manuscript, the candidate kinases were enriched from the motif extraction of the phosphosites. Were these kinases of interests also enriched in the BMDMs ? In the example of TAK1, it is upstream regulator of MAPK pathway and was enriched in the myeloid cells, yet the result showed that the MAPK pathway was not significantly altered in the BMDMs (Extended Data Fig. 1J). This comparison raised the concern whether the results presented in this manuscript are cell type-dependent. A systematic comparison on the putative kinases in these two cell lines from the motif analysis has to be done to clarify this point.

6. The results of the subcellular localization of the dynamic TNF-mediated phosphorylation events are quite interesting. The data show that many proteins have dual localization in the basal and treated condition. There are documented examples that phosphosite will regulate the protein subcellular localization. One inherent question: will this localization depend on phosphorylation (site-dependent) ? How to link the proteome-based localization data to infer the potential regulation of localization from individual phosphoprotein of different site ?

Though the Figure 3 shows the summary of overall change, the relationship between subcellular localization of the individual proteins at basal level, TNF activation and TAK1 is missing. Presentation of selected examples, such as TNIP1, to show both temporal change and subcellular distribution will be helpful. TNIP1 was detected residing in the cell membrane after stimulation. What is the localization at basal level ? After the TAK1 inhibition, was it reallocated or stay in the basal localization ? Such data will be interesting to show the two dimensional map of phosphorylation-mediated signaling in time and space.

7. Compared to previous literature, what are the common findings? Any consistent evidences on the TNF signaling pathways? The Authors may further discuss the advancement of this study.

Minor comments:

The Extended Data figure legends need to be presented on the same page as the figures

Friday, July 02, 2021

Point by point answers to reviews of **“Phosphoproteome profiling uncovers a key role for CDKs in TNF signaling”** by Maria C Tanzer, Isabell Bludau, Che A Stafford, Veit Hornung and Matthias Mann

We were pleased by all three reviewers' positive assessment of our manuscript. In this revision we have performed extensive proteomic and biological experiments. However, after discussion with the editor, we have limited the scope to what is reasonable to expect in a single study. In particular, this means that we could not be expected to confirm that newly identified phosphorylation events are relevant for cellular signaling in detailed functional follow up studies. That said, we have performed very extensive proteomic and biological experiments to address all concerns of the reviewers.

One of the major concerns about the manuscript was the reproducibility of the TNF-regulated phosphorylation events in different cell lines. By performing phosphoenrichment experiments in three additional cell lines originating from different tissue (A549, HT29, U2OS) we now show that the majority of significantly upregulated phosphorylation events in U937 cells are also upregulated in all other cell lines tested. Additional enrichment and kinase motif analysis revealed regulation of phosphorylation events on many overlapping proteins between human and murine cell lines. This demonstrates that the regulated phosphorylation events reported or not only specific to the U937 cell line. Importantly, all three human cell lines were also sensitive to the synergistic cell death of TNF-mediated apoptosis inducers (TNF + SM or TNF + TAK1i) in combination with various CDK inhibitors, including CDK12/13 inhibitors. Thus our finding of CDK12/13 dependent upregulation of TNF-target genes also holds true for other human cell lines.

All technical and minor concerns were fully addressed and all raw data of bar graphs, scatter plots and averaged heatmaps and uncropped western blots are provided in the source data file.

Reviewer #1 (Remarks to the Author):

Tanzer and colleagues provide a rich proteomic resource for scientists interested in TNF signaling in myeloid cells. In the latter part of the paper, they focus on transcriptional CDKs, which were activated by TNF and mediated phosphorylation of a subunit of RNA Pol II. Activation of pro-survival genes in this manner is proposed to limit TNF-induced cell death. Experiments are well controlled and I support publication. I have only a few suggestions for the authors to consider below.

We thank the reviewer for this very positive assessment of our manuscript and we indeed hope to have provided a rich and unprecedented resource for the large TNF community.

Specific comments:

1) The authors highlight signaling downstream of TNFR1, but can they link any of the phosphorylation events to NIK activation downstream of TNFR2?

This is an interesting point. We did not find any phosphorylation of NIK (MAP3K14), but detected significantly increased phosphorylation of NFKB2 (S858 and S872) at 15 minutes of TNF stimulation (see below). NFKB2 is activated by IKK α downstream of NIK. The phosphorylation of NFKB2 at S872 has indeed been found to be mediated by IKK α (Xiao et al, JBC, 2004, PMID: 15140882). Surprisingly, however, our data shows that both phosphorylations are inhibited by the TAK1 inhibitor 7-Oxozeaenol (see below). As this inhibitor does not target NIK directly, TAK1 is either involved in NIK or IKK α activation (Ninomiya-Tsuji et al, JBC, PMID: 12624112).

Z-scored phosphosite intensities on NFKB2 in TNF-stimulated U937 cells for 15 minutes (n = 4, \pm SD). Empty circles represent imputed values.

2) Fig 2a suggests that activation of TBK1/IKK ϵ in response to TNF is divorced from activation of TAK1 and IKK1/2. This differs from the model of Philip Cohen’s group in their 2011 paper (PMID: 21138416), which states in the abstract “the TNF α -induced activation of the IKK-related kinases is mediated solely by the canonical IKKs.” Perhaps their results differ because they used MEFs rather than myeloid cells? Can the authors comment on whether they think their results in myeloid cells are more broadly applicable to other types of cells.

We thank the reviewer for sharing this information. Our data clearly show that 7-Oxozeaenol, the TAK1 inhibitor, and TPCA-1, the IKK inhibitor, are not able to inhibit certain TNF-induced phosphorylations, which can be reduced by the TBK1/IKK ϵ inhibitor MRT67307. Phosphorylation of CYLD, for example, is known to be mediated by IKK ϵ and we cannot detect any inhibition upon 7-Oxozeaenol and TPCA-1 treatment. As the reviewer already stated, the experiments of that specific paper were done in a different cell system, derived from a different species. This could explain the differences we observe. However, we also detected a strong inhibition of a phosphorylation event on TBKBP1 upon TPCA-1 treatment (Fig. 2e), which is required for certain functions of TBK1 (e.g. growth factor signaling, Zhou et al 2019 Nat. Cell. Biol. PMID: 31792381). Therefore, we conclude that canonical IKKs will most likely regulate the activity of non-canonical IKKs, but this does not influence the phosphorylation events we detected around the LUBAC complex upon TNF stimulations.

We had chosen U937 cells as they are of myeloid origin and myeloid cells are one of the most relevant and responsive cells to TNF stimulation. To answer the reviewer’s question if our results from myeloid cells are more broadly applicable to other types of cells, we have now performed phosphoenrichment experiments in HT29 (colon carcinoma cell line), A549 (lung epithelial cell line) and U2OS (osteosarcoma cell line) treated for 15 minutes with TNF (see also response to point 1 of reviewer 2). The results reveal

that the majority of significantly upregulated phosphorylation events in U937 are also upregulated in all other cell types tested, while also revealing regulation of cell type specific phosphorylation events.

3) Fig 6 and EDF. 6 focus on regulation of cFLIP as a potential mechanism by which the CDKs suppress cell death. What about the regulation of other highly induced genes such as A20/TNFAIP3, which suppresses the transition from TNFR1 complex 1 to death-inducing complex II?

Indeed, other proteins involved in cell death downstream of TNF could be affected. MCL-1 levels are known to be regulated by CDK9 reduction, but remained the same upon CDK12/13 inhibition (see below) (Lemke et al, CDD, PMID: 24362439). Also, A20 induction upon TNF was slightly reduced by the pan-CDK inhibitor Dinaciclib, while the CDK12/13 inhibitor, THZ531, had no effect (see below). This indicates that inhibition of different transcriptional CDKs has differential impact on pro-survival proteins. We cannot exclude that other proteins, that we have not tested, are affected and contribute to cell death induction upon CDK12/13 inhibition and TNF stimulation. However, in our data cFLIP, a direct inhibitor of caspase-8, is the most regulated and likely candidate to mediate this cell death. We added this information to the Extended Data Fig. 5.

Western blots of U937 cells treated with TNF alone or in combination with the panCDK inhibitor Dinaciclib (6 nM) or the CDK12/13 inhibitor THZ531 (200 nM) stained for A20, MCL1 and Actin levels.

Reviewer #2 (Remarks to the Author):

Tanzer et al. provide a thorough mass spec analysis revealing a large number phosphorylation sites in TNF-stimulated U937 cells. The results are of potentially high interest to scientists who study TNF signaling. The study also examines phosphorylation events that depend on the activity of transcriptionally acting CDKs, further broadening the relevance of some of the identified phosphorylation sites. The amount of generated data is impressive, however, the mass spec work is mostly based on the analysis of one cell line, U937 cells, in which the authors identify a number of novel phosphorylation sites (figs. 1-4). It seems that most of the phosphorylation sites detected in U937 cells cannot be detected in BMDMs or MEFs. Are the findings obtained in U937 cells which are reported in figs. 1-4 representative of TNF signaling in general or could they be specific to this cell line? By the end of figure 4 the authors have essentially

provided a list of several potentially interesting phosphorylation events that can now be validated functionally in U937 cells and in other cells. However, such a validation is not presented. The only attempt at validation is presented in extended figure 4b and shows no significant effect of XIAP phosphorylation sites on TNF induced cell death. It also remains somewhat unclear how this validation attempt was conducted. How were the XIAP KO cells made? Has this been done by CRISPR/CAS9 and how were the KO cells reconstituted? What are the levels of the reconstitution that were achieved with the individual mutants and how different are these levels to reconstitution with wild type and the expression level in the parental cell line before KO? Thus the first part of the study offers many potentially meaningful phosphorylation sites but none of them is adequately validated functionally with respect to TNF signaling. The study next embarks on a different direction and examines the regulation of TNF target gene expression mediated by transcriptional CDKs. This part is biochemically explored in more detail, yet the knowledge of transcriptional CDK inhibition and sensitization to cell death induction by TNF superfamily members is not novel and has been reported in several other studies before. The authors reference these studies appropriately but their work fails to provide an advance beyond the current understanding of this process.

We thank the reviewer for this thorough review and for pointing out the potential usefulness of our work for the community. Regarding the central point about extending and validating our results in other systems, we have now added three more cell lines (see above and below). We also address the specific points raised above in the detailed questions below. For functional validation, please see our answer in the summary above.

The following major points need to be addressed by the authors:

1. Figs. 1-4 show results on TNF stimulation of U937 cells. Are any of the newly identified phosphorylation sites also observed in other cell types or cell lines, including in mouse cells? A biochemical and functional validation of specific phosphorylation events on a few of the newly identified targets would provide for a useful validation of the results obtained by mass spectrometry.

We agree with the reviewer that it is important to show that the phosphorylation sites we detected are not only restricted to U937 cells. Therefore, we additionally stimulated the human A549 (adenocarcinomic alveolar basal epithelial cell line), HT29 (colorectal adenocarcinoma cell line) and U2OS (osteosarcoma cell line) cells with TNF (15 m) and performed a phosphoenrichment experiment. When we filtered for significantly upregulated phosphosites (Student's sample t-test, FDR < 0.05; Student's t-test difference (Log2) > 3) in U937 cells and generated a heatmap to demonstrate the regulation of these phosphosites in the different human cell lines, we now demonstrate that the majority of these site is also upregulated in other cell lines (see below). Some phosphorylation events could not be detected in other cell lines, most likely due to protein abundance differences. We added the tables containing the t-test results of the different cell lines as Supplementary Table 1 to the manuscript.

Z-scored intensities of phosphorylated peptides which are significantly changing in TNF-stimulated U937 cells compared to untreated cells (Student's sample t-test, FDR < 0.05; Student's t-test difference > 3) and corresponding z-scored intensities in A549, HT29 and U2OS cell lines.

While it is difficult to systematically compare human phosphosites with murine phosphosites due to differences in protein length and sequence, we can compare the proteins whose phosphorylation is significantly regulated upon TNF stimulation. Thereby we found that out of 449 proteins significantly phosphorylated or de-phosphorylated upon TNF treatment in U937 cells (kinase inhibitor dataset) 318 of the corresponding murine proteins were also found to be significantly phosphorylated or de-phosphorylated in BMDMs. This indicates that the large majority of proteins regulated on the phosphorylation level in human cells are also regulated in murine BMDMs and points to a remarkable congruence over 100 million years of evolutionary history.

Many of the regulated phosphorylation events in our datasets have already been shown to impact TNF signaling. To identify novel phosphorylation events important for TNF signaling we tested next to XIAP phosphosites other phosphosites for their functionality in TNF-induced cell death RIPK3 S410 (reconstituted in RIPK3 knock out U937 cells) and IKBKB S675 S697 (reconstituted in IKBKB heterozygous U937 cells, see below). We could not detect differences between mutated versions, which prevent phosphorylation to the wild-type counterpart. We speculate that they may have a function in a specific in vivo setting at endogenous levels. This is, however, almost impossible to determine and would be beyond the scope of this resource study. We stress that many of the regulated phosphorylation events in our datasets have been shown to impact TNF signaling, including phosphorylations on CYLD, MAPK14, NFKB1, MAPK3, ATF2 and more.

Propidium iodide staining of U937 cells deficient for RIPK3 and IKK β respectively expressing wt and phosphoablating RIPK3 and IKK β variants upon addition of 1 μ g/ml doxycycline 24 h before treatment as indicated with hTNF (100 ng/ml) and Smac mimetic (1.25 μ M) and Idun-6556 (10 μ M) for 24 h to induce apoptosis and necroptosis. Western blot of the cells were conducted to test for IKK β expression upon doxycycline addition.

To further answer the reviewer's concern, we next took a bioinformatics approach. Applying the functional prediction score of human phosphosites published by Ochoa et al (Nat. Biotech., 2020, PMID: 31819260) based on 59 features, a number of identified phosphosites in our dataset that are significantly changing, are predicted to have a function (see below). This is clearly in line with the fact that many of our regulated sites are already known to be functionally important.

Functional score based on Ochoa et al of significantly changing phosphosites (time course data set). Phosphosites with known regulatory function have an average score of 0.55.

2. In several instances the authors use kinase inhibitors which are often only selective and not specific. Have the authors compared them to loss of expression of the main targets of such inhibitors?

We thank the reviewer for these suggestions. In the previous manuscript, we had only looked at the phosphorylation level of the kinases of interest or their direct substrates, but had not analyzed the protein levels of potential other targets. Prompted by the review, we now performed a proteome analysis of the same samples used for the phosphoenrichment. This revealed that overall only 37 proteins were significantly regulated between different inhibitor treatments and the untreated control (Student's t-test, FDR < 0.05). Amongst these 37 proteins were 5 kinases (shown in bold), which do not regulate TNF signaling. We therefore conclude that these inhibitors do not influence our functional phosphoproteomics results and interpretations.

z-scored intensities of proteins which are significantly changing (Student's sample t-test, FDR < 0.05) upon at least one of the inhibitor treatments compared to the respective DMSO treated (UT) control.

3. One of the aims of the study is to evaluate how transcriptional CDKs are regulated by phosphorylation. The authors should therefore validate the reported findings using suitable systems. How does mutation of the main phosphorylation sites identified for transcriptional CDKs affect TNF signaling? Additionally, the authors should repeat at least some of the phosphoproteomic experiments in the presence of the CDK inhibitors. In figure 5 they only provide a proteomic analysis of TNF-treated U937 cells in the presence or absence of the inhibitors and the data provided in Extended Data Fig. 5a cannot be compared with the phosphoproteomic analysis carried out in figure 5a, not only because the authors only show selective changes in Extended Data Fig. 5a, but also, and most importantly, because the phosphorylation sites identified upon TNF stimulation are not consistent between the two figures. The authors should carry out a wider screen using different cell types to evaluate how reproducible the observed phosphorylation events are since the comparison between U937 and BMDM already resulted in different outcomes. As an example, TNF stimulation induced a different pattern of phosphorylation of transcriptional CDKs in BMDMs versus U937 cells (comparison between figs. 4e and 4f), and in BMDMs no phosphorylation events were detected for CDK9 upon TNF stimulation, yet there was a strong upregulation of phosphorylation of CDK14, whose role in transcription is not well characterized. The authors do not comment on this result but instead mainly focus on the effects observed in U937 cells. How the authors explain the discrepancy between these two cellular systems? Are the observed effects cell-type specific?

We agree with the reviewer that a more thorough analysis of CDK regulations should be undertaken as far as possible and practical in a single study. In this revision, we now investigate the regulation of CDKs

in HT29 and U2OS cells during TNF stimulation and TNF-induced apoptosis. A bioinformatic enrichment analysis using Fisher exact test's reveals that transcriptional CDKs are also one of the most strongly regulated kinases in these two cell lines upon TNF stimulation (A). Thus regulation of CDKs is definitely as shared and generalizable biological feature. At the site level, while several of their phosphosites are overlapping, many seem to be regulated in a cell type specific manner. This indicates that regulation of certain CDK phosphorylation events seem to be cell type dependent (B, C). Detection of CDK12 activating phosphorylation in BMDMs already demonstrates its activation during TNF stimulation. Further, functional analysis of other phosphorylation changes by generation of phosphomimetic and phosphoablating mutations is difficult considering the complexity of their regulation. Therefore, we had decided to proceed with specific and selective inhibitors targeting CDK12/13. Our new phosphoproteomics experiments, show that these inhibitors have a comparable impact to the IAP inhibitor Birinapant on TNF regulated phosphorylation events (see below) and induce synergistic cell death with TNF in different cell lines tested (see response to point 8).

A) Fisher's exact test of kinases significantly regulated via phosphorylation upon apoptosis induction (TNF and Smac mimetic Compound A) of HT29 and U2OS cells (FDR < 0.05). Green indicates TNF and SM treatment. B, C) Heatmap of means of z-scored phosphosite intensities of phosphosites significantly changing upon TNF treatment for 3 h (FDR < 0.05) in HT29 and U2OS cells. Cells were pretreated for 30 m with Compound A (5 uM). Red indicates TNF treatment. Green indicates TNF and SM treatment.

Heatmap of z-scored phosphosite intensities of phosphosites significantly changing upon TNF treatment for 3 h (FDR < 0.05) in U937. U937 were pretreated for 30 m with Birinapant (1.25 uM), Dinaciclib (6 nM) and THZ-531 (200 nM).

4. In fig. 5 the authors investigated if and how CDK inhibitors affect TNF mediated regulation of target genes at the transcriptional level. It is not surprising that Dinaciclib, CDK12-IN3 and THZ531 reduce transcription of TNF target genes, as it is known that CDK9, 12 and 13 regulate transcriptional elongation so that their inhibition would affect productive transcription, including the transcription of TNF target genes. Furthermore, the authors only show a reduction in the levels of MCP-1 mRNA in U937 cells and of IP10 release for BMDMs (Extended Data Fig. 5a) upon stimulation with TNF in the presence of the different CDK inhibitors. It is unclear why the authors chose to detect and compare two different cytokines which makes it difficult to assess the impact of CDK inhibitors on TNF-induced cytokine production, especially because from the data provided it seems that the different CDK inhibitors exert different regulatory functions in the two cellular systems employed. Additionally, fig 5e-h and fig 6 lack novelty. As correctly cited by the authors, it has already been shown that inhibition of the transcriptional CDK9 affects the expression level of the anti-apoptotic protein FLIP and that it was therefore expected that TNF-induced FLIP upregulation can be prevented by inhibition of transcriptional CDKs and that this would enhance TNF induced cell death.

We apologize for our inconsistency regarding our choice of target cytokines/chemokines in the different cell systems. This choice was made out of convenience, as we already had IP10 ELISA plates and MCP-1 qPCR primers and based on prior knowledge about the upregulation of these target genes upon TNF treatment. We now included the mRNA levels of IP-10 in U937 cells, which should make it easier to assess the impact of CDK inhibitors on TNF-induced cytokine production in the different cell systems (see below and Extended Data Figure 5). In both, U937 cells and BMDMs we see reduced expression of cytokines upon CDK12/CDK13 inhibition even though the reduction efficiency varies.

Fold change of IP-10 mRNA levels in U937 cells treated for 4 hours with TNF alone or in combination with various CDK inhibitors.

Indeed, the regulation of FLIP expression has been shown before to be regulated by transcriptional CDK9. However, we show here for the first time that this is also the case for CDK12 inhibitors downstream of TNF. We believe this is an important finding as CDK9 and CDK12 inhibitors have a different impact on cells, also shown by unique upregulation of FLIPs upon CDK12 inhibitor treatment and the differential potency to synergistically induce necroptosis (CDK9 inhibition exacerbates necroptosis to a much higher extent than CDK12 inhibition).

5. The authors state “Our data thus suggests a potential association of these proteins with the LUBAC complex upon TNF signaling” (page 7, line 30). However, the phosphorylation of ARFGAP2, IQSEC1 and FERMT3 are suppressed by MRT67307 treatment prior to TNF stimulation. Therefore these results do not show a direct link to LUBAC as these proteins could also interact with another component of the TNF receptor complex I or form part of downstream signaling pathways influenced by TBK1/IKKe.

The reviewer state that ‘the phosphorylation of ARFGAP2, IQSEC1 and FERMT3 are suppressed by MRT67307 treatment prior to TNF stimulation’, however, this is not the case in our data or our text. In fact we did not see inhibition of ARFGAP2, IQSEC1 and FERMT3 phosphorylations in MRT67307 treated cells prior to TNF stimulation compared to the untreated control (cells without inhibitor) as we did not detect their phosphorylated peptides in these control conditions. This indicates that these phosphorylation events are induced by TNF and inhibited by the TBK1/IKKe inhibitor. Our visualization in Fig. 2 may have been misleading (grey was supposed to be non-detected) and we have now fixed this in the revision.

In any case, we completely agree that our results do not prove a direct interaction of these specific phosphorylated proteins with the LUBAC complex. However, we speculate on a potential interaction as phosphorylations of these proteins are regulated similarly to proteins that have been shown to be located at the LUBAC complex.

6. In Extended Data Figure 4 the authors went on to identify the different phosphorylation sites on known components of the TNF signaling pathway. Here they mainly focus on the identification and validation of XIAP phosphorylation sites by knockout and reconstitution with phospho-ablative or phospho-mimetic approaches. However, it is difficult to assess the results as neither data on the efficiency of XIAP knockout in U937 nor detailed explanation of the methodology used to achieve the XIAP knockout and

reconstitution are provided in the materials and methods section. The authors should assess this effect in a different cellular system perhaps in cells of epithelial origin.

We apologize for this lack of detail regarding the description of XIAP knockouts and the knockout efficiency. The XIAP knockouts were generated using two CRISPR-Cas9 RNPs (gRNA1: TTCTCTTTTAGAAAAGG; gRNA2: GACTTTTAACAGTTTTGA). CRISPR-Cas9 RNPs were assembled by annealing synthetic, chemically stabilized crRNA:tracrRNA pairs (IDT) at 95 °C for 5 min followed by incubation at room temperature for 30 min. gRNAs were then mixed with recombinant NLS-Cas9 protein for 20 min at room temperature. Cas9 (40 pmol) was added for each 100 pmol of XIAP gRNA. RNPs were then mixed with the cell suspension and nucleofection was conducted using the SG Cell Line 4D-Nucleofector X Kit S (Lonza) on the X-unit of a 4D-nucleofector (Lonza) (program FF-100). After nucleofection, cells were collected from the nucleofection cuvettes and transferred into a 96-well plate. After 48 hours U937 cells were then subject to single cell dilution cloning in a round-bottom 96-well plate. When colonies became visible after approximately 3 weeks, clones were collected and subjected to MiSeq analysis as previously described (Schmid-Burgk, Genome Res, 2014, PMID: 25186908). We also show the sequencing results below. This information has been added to the Methods section in the revised manuscript.

Please find the Western blot analyses demonstrating XIAP deficiency and reconstitution of the serine to alanine mutants below. The protocol for the generation of knock outs is now added to the manuscript. We followed the reviewer’s suggestion to repeat this experiment in HT29 cell lines, a colon cancer cell line with epithelial morphology. However, this did not lead to a detectable increase in cell death in XIAP heterozygous cells or decreased cell death in XIAP variants overexpressing cells (see below).

Propidium iodide staining of HT29 cells expressing wt and phosphomimetic (on top) and phosphoablating (below XIAP variants upon addition of 1 μ g/ml doxycycline in heterogeneous XIAP knock out cells (CRISPR/Cas9 pool) 24 h before treatment with hTNF (100 ng/ml) and Smac mimetic (1.25 μ M) to induce apoptosis for 24 hours. Western blot of the same cells were conducted to test for XIAP variant expression upon doxycycline addition.

Western blot of wild-type U937 cells and a XIAP deficient U937 clones, which were reconstituted with wild-type and mutated XIAP. Addition of Doxycycline (1 μ g/ml) for 24h induced expression of XIAP variants. This Western blot is a new supplementary figure in the revised manuscript (Extended Data Fig. 4)

The Sanger ICE analysis (<https://ice.synthego.com>) sequencing results demonstrate the XIAP knock out efficiency of the two clones we used for the reconstitution experiments.

7. Based on the results provided in fig. 4g-i the authors claim that TNF stimulation induces increased CDK9 phosphorylation which in turn correlates with increased phosphorylation of the CTD of RPB1 at Ser-2, and this could be prevented by the different CDK inhibitors employed. It seems surprising however that at time point “0” there was no detectable Ser-2 phosphorylation of the CTD of RPB1 whereas in the same condition the authors were able to detect the activating phosphorylation of CDK9 (pCDK9) at position T186. Moreover, inhibition of CDK9 with Dinaciclib or AZD4573 induced an increase in the levels of pCDK9 independently of TNF stimulation. How the authors explain this and what happens to pCDK9 upon treatment with the CDK12 inhibitor THZ531?

We thank the reviewer for pointing out these discrepancies. Indeed, similar to the phosphoproteome analysis, we also detected moderate levels of phosphorylated CDK9 (T186) in the untreated condition of the Western Blot (Figure 4g, i). These lower levels might not suffice to trigger RPB phosphorylation, or the phosphorylation of RPB1 could be localized at a different position than S2. To our knowledge phosphorylation of the CTD of RPB1 is highly regulated and depends on many factors such as various CDKs.

The increased levels of phosphorylated CDK9 upon Dinaciclib and AZD4573 treatment in Figure 4g, i are likely due to the fact that inhibitors do not always lead to reduced phosphorylation levels but can lock the phosphorylated target in an inactive position. The phospho CDK9 (T186) levels did not change upon THZ531 treatment (see below).

Western blot of U937 cells treated with TNF alone or in combination with THZ531 (200 nM) stained with antibodies against phosphorylated CDK9 and Actin.

8. In fig. 6 the authors show that transcriptional CDK inhibitors synergize with TNF in U937 but do not discuss that also independent of TNF and SM the combination of SM with IDN-6556 results in a synergistic cell death induction in U937. The authors should clarify this. Importantly, in the text the authors stated that “This synergistic cell death is not restricted to U937 cells but also present in mouse dermal fibroblasts (MDFs) and murine BMDMs” (page 18, line 24). However, there are no results in the present study supporting this statement as the data provided in fig. 6a-i shows a different pattern of sensitivity for the treatment with TNF/CDK inhibitors in the cellular systems employed since mouse dermal fibroblasts do not show this strong sensitivity to TNF/CDK inhibition treatment observed in U937 cells. How the authors explain these differences? Are the synergistic effects observed only specific for U937 cells?

We apologize if our text was misleading. We amended the text (marked in yellow in the revision) to make the message clearer. The sentence quoted by the reviewer did not refer to the synergistic cell death mediated by CDK inhibitors and TNF treatment, but is related to the synergistic impact of CDK inhibitors and the apoptosis inducers TNF and Smac mimetics. As the reviewer correctly observed we see strong synergistic cell death of the combination of TNF and CDK inhibitors only with U937 cells. This is however not due to a unique feature of U937 cells but it is rather due to the fact that TNF treatment on its own induces low levels of cell death in this cell line. We conclude in the manuscript that CDK inhibitors at the concentrations we are using are poor inducers of cell death and rather exacerbate existing cell death. We changed the text to convey the message accordingly. Importantly, we observed a synergistic effect of TNF and especially TNF and SM in other cell lines including HT29, U2OS and A549 cells (see below). This data is now added to Extended Data Fig. 6.

Bar graphs demonstrating cell death via propidium iodide staining of A549, HT29 and U2OS cells treated with TNF, TNF and the Smac mimetic Birinapant, TNF and Smac mimetic (Compound A) and TNF and TAK1 inhibitor with or without the pan-CDK inhibitor Dinaciclib (12 nM), CDK12/13 inhibitor THZ531 (200 and 400 nM) and CDK12 inhibitor SR4825 (120 nM). This is now added to Extended Data Fig. 6.

Minor points which should be addressed are:

- In the legend of fig. 5 the authors should mention the concentration of AZD4573 used.
- The statement “IKKε phosphorylates and activates the deubiquitinase CYLD” is not correct. The authors refer to Huttu et al. (2009) who published that IKKε can phosphorylate CYLD at Ser418 and thereby decrease its activity.

We thank the reviewer for the suggestion and correction. We amended the text accordingly. Please find the changes highlighted in yellow.

Reviewer #3 (Remarks to the Author):

In the paper by Tanzer et al. the authors applied quantitative phosphoproteomics to dissect the dynamic and subcellular phosphoproteomic alteration during the course of TNF stimulation in myeloid cells. Motif enrichment analysis was performed for these altered phosphorylation event to identify upstream kinases. Compared to previous studies on TNF signaling, this study achieved much enhanced phosphoproteomic coverage to uncover the systems map of the TNF signaling. A very interesting finding inferred from the phosphoproteomics profiling is the discovery of the role of CDK kinase activity in TNF-induced cell death. Two cell types were studied in both phosphoproteomics and cell biology experiments, yet some of the results presented seem to be different, which raised the concern whether the findings are general in TNF signaling and not cell type-dependent. Nevertheless, this study provides rich resource to show the two dimensional systems map of phosphorylation-mediated TNF signaling

in time and subcellular space. Given the clinical significance of CDK family and TNF signaling in inflammation and cancer therapy, these findings will be of significant interests. However, a number of significant issues require clarification to further improve the manuscript.

We thank the reviewer for the positive feedback. The comments were very helpful to make our manuscript clearer and more comprehensible.

1. The use of data-independent acquisition (DIA) provided good overall phosphoproteomics profiling in this work. DIA is known to be a more reliable method compared to conventional approach. How is the reproducibility in replicate runs and overlap between different time point? The Authors have to describe the quantification results of each time point.

Indeed, the data-independent acquisition mode reduces missing values and results in better reproducibility of replicate runs also for phosphoproteomic experiments. Additionally, DIA enables us to identify and quantify more phosphopeptides compared to DDA as has been described on Orbitrap instruments before (Bekker-Jensen et al, 2020, Nat. Comm., PMID: 32034161). Still, our study is one of the first of its kind to use DIA. As you can see below, based on Pearson correlation the reproducibility of our runs is very good. The identification results (mean of each replicate) and the quantification results of each time point are shown in the heat map below. All t-tests are available through the web browser in our TNF viewer web site (www.tnfviewer.biochem.mpg.de).

Average phosphosite numbers including multiplicities:

UT_15sec	11952
UT_30sec	12074
UT_60sec	11768
UT_90sec	11972
UT_3min	12274
UT_5min	11911
UT_8min	11968
UT_15min	12057
UT_60min	11423
TNF_15sec	11917
TNF_30sec	11888
TNF_60sec	12157
TNF_90sec	11934
TNF_3min	12290
TNF_5min	12132
TNF_8min	12394
TNF_15min	12391
TNF_60min	11840

Phosphoproteome correlation matrix of untreated and TNF-treated samples (Pearson correlation). The scatter plot represents the correlation between two replicates. The correlation matrix is now added to Extended Data Fig. 1.

2. While it is understood that the phosphopeptide identification may generate multiple phosphorylation sites per peptide, can the Author explained more clearly on how the 60,000 phosphopeptides map to 28,000 class 1 sites ? I have one major concern on the assignments of phosphosites with different sites in the nearby location. In all of the heatmap results (eg, Fig. 2 e-f, Extended Data Fig. 2C, Fig. 3g-l, Fig. 4), most of the proteins with list of multiple sites on the same peptides have identical intensity in the heatmap. It is less likely that the nearby sites have identical alteration. Are these from the same peptides with few phosphosites which cannot be distinguished for the exact phosphosites? Such condition has to

be differentiated from the one that has unambiguously localized phosphosites on the same peptides. The identification results and figures in this manuscript have to be clarified and revised.

We agree and apologize that we did not explain the summary statistic of the phosphosites sufficiently. We added a description in the method section (highlighted in yellow). The 60,000 phosphopeptides were detected in all experiments combined, while we detected 28,000 class 1 phosphosites in the time course experiment (figure 1).

As the reviewer correctly observed, different phosphosites can have the same intensities if they are localized on the same peptide. This is not due to ambiguous localization but to the localization of the same phosphosites on the same peptide. The Perseus Plugin we used collapses phosphoions to phosphosites. Importantly, it does not sum up the intensities of a phosphosite on peptides, if different phosphorylations are also present. To illustrate, the intensity of MAPK14_T180_M1 and MAPK14_T180_M2 (for MAPK14_T180_Y182) are different in our results. For each phosphosite on a multiple phosphorylated peptide we receive a row with the same intensities as these phosphorylations are localized on the same peptide. While MAPK14_T180_M1 represents the singly phosphorylated peptide, MAPK14_T180_M2 and MAPK14_Y182_M2 share the same intensity as they represent the two phosphosites on the same peptide. Different phosphosites on the same peptide can have slightly different fold changes due to imputation. Also, each collapse key (gene_position_multiplicity) is unique, which means that if a phosphosite is present on two peptides that carry a different phosphosite, just one row will be assigned.

Phosphosites located on phosphopeptides with more than two phosphorylations are labelled with a multiplicity of 3 (M3). We set a PTM localization probability filter of > 0.75 .

We have now explained this more clearly in the Me

3. The regulation of a phosphosites may be contributed from either the protein expression or phosphorylation degree. For the quantitation result shown in this study, were they normalized to the change of protein expression?

We agree with the reviewer that phosphorylation changes can sometimes occur due to changes of the respective proteins. However, due to the early time point (15 min) we did not expect many significant protein level changes and therefore did not normalized our phosphodata. Based on the reviewer's question we tested by comparing the proteome of untreated cells with cells treated with TNF for 15 minutes (see left). In a first repetition this led to only two significantly changing proteins and in a second experiment there were no significant changes.

Also at later time points, such as in the cell death induction experiments, the phosphoproteome data always dominated the respective proteome data (Extended Data Fig. 4l, j, k). Thus, for biological interpretation, normalization was not necessary.

Scatter plot of Student's t-test of the proteome of TNF treated U937 cells for 15 min compared to the untreated control. Significant changes are highlighted in red (FDR < 0.05).

4. The dynamic phosphoproteomic profiles at 15 min of TNF treatment was further analyzed in the murine bone marrow derived macrophages (BMDMs), in which the phenotypic nature is different than the myeloid cells. The immune response-related functional terms enriched from both cell systems are different. In the validation results of TNF-induced cell death, the Authors only mentioned that the experiment was also performed in BMDM cells. No further elaboration was made to link the result in the selected examples in the U937 cells. Whether the result from BMDM cells provide additional insight is not clear. The Authors have to clearly justify the rationale of using these two cell types. To dissect the common and unique TNF-induced cascades, comparison on the temporal response of both cell types at 15 minutes and examined whether these altered expression can be restored at later time point will be critical information.

We stress that we chose to investigate phosphorylation changes in BMDMs as they are a primary model of myeloid cells. While we would expect differences in kinase levels and subsequent phosphorylation events between transformed and primary cells, in bioinformatic enrichment analysis of differentially phosphorylated proteins upon TNF stimulation many enrichment terms are shared or involve many of the same proteins between U937 and BMDMs (see Extended Data Fig 1). While it is difficult to systematically compare human phosphosites with murine phosphosites due to differences in protein length and sequence, we can compare the proteins whose phosphorylation is significantly regulated upon TNF stimulation. As pointed out for reviewer 2, we found that out of 449 proteins significantly phosphorylated or de-phosphorylated proteins upon TNF treatment in U937 cells (15 m TNF stimulation, kinase inhibitor dataset) a remarkable 318 of the corresponding murine proteins were also found to be significantly phosphorylated or de-phosphorylated in BMDMs. This shows that many of the same proteins in U937 cells and BMDMs are regulated during TNF treatment. Furthermore, kinase motif enrichment analysis revealed enrichment of the same motifs between stimulated U937 cells and BMDMs (see response to point 5).

We did not focus on the cell death data of BMDMs and the comparison to U937 cells as cell death induction of BMDMs was achieved by inhibition of TAK1 inhibitors and was therefore different to the induction of U937s. We chose to use the TAK1 inhibitor to induce cell death in BMDMs due to their very low sensitivity to Smac-mimetics. Importantly, our results demonstrate the regulations of the classical members of the TNF signaling pathway e.g. phosphorylation of RIPK1 and RIPK3, as they are in contrast to U937 cells highly expressed in BMDMs and therefore more readily detectable. We also described the regulation of proteins involved in RNA processing in both, U937 cells and BMDMs (Fig. 4c; Extended Data Fig. 4g, h, i). Additionally, we detected increased phosphorylations of ATM kinase substrate motifs in U937 cells and BMDMs (Fig. 4c; Extended Data Fig. 4g). We also demonstrated that regulation of CDK phosphorylation during TNF not only occurs in U937 cells but also in BMDMs (Fig. 4f).

When comparing early (15 min) and late (1h and 3h for U937 and 1h for BMDMs) time points of TNF stimulation, we detected activation of innate immune pathways like RIGI, Myd88, Interleukin1, MAPK signaling at early but not at late time points in both U937 and BMDMs (see below). In BMDMs we detected increased phosphorylation of proteins involved in Rac, Rho GTPase and Actin binding and cell migration at early and late time points. In U937 cells we also detected regulation of proteins involved in migration at 1 h of TNF treatment. Proteins involved in the activation of the RNA polymerase II promoter, hence regulating transcription, are increasingly phosphorylated at 1h and 3h of stimulation. In U937 TNF treatment alone induces low levels of apoptosis at 3 h and therefore many of the terms enriched during apoptosis are also enriched upon TNF treatment. Therefore, we conclude that the dominant early

signature of protein phosphorylation involved in innate immunity is downregulated at later time points, when proteins involved in migration and transcription are primarily regulated.

BMDMs

U937

Fisher's exact test of proteins significantly phosphorylated upon TNF stimulation (15 m, 1h, 3 h) in BMDMs (top) and U937 cells (bottom) (p-value < 0.02).

5. In this manuscript, the candidate kinases were enriched from the motif extraction of the phosphosites. Were these kinases of interests also enriched in the BMDMs ? In the example of TAK1, it is upstream regulator of MAPK pathway and was enriched in the myeloid cells, yet the result showed that the MAPK pathway was not significantly altered in the BMDMs (Extended Data Fig. 1J). This comparison raised the concern whether the results presented in this manuscript are cell type-dependent. A systematic comparison on the putative kinases in these two cell lines from the motif analysis has to be done to clarify this point.

It is known that TNF signaling activates TAK1-dependent NFkB and MAPK signaling in U937 cells and BMDMs. TAK1 inhibition - for example in combination with TNF - induces cell death in both cell types (see Figure 2 b and Sanjo et al, J Immunol., PMID: 31243089). In extended Figure 1 we primarily focused on the enrichment terms that demonstrate phosphorylation of proteins involved in other innate immune

pathways and left out the terms related to MAPK in the BMDM dataset. However, the analysis also retrieved the term ‘MAPK signaling pathway’ as significantly enriched in TNF treated BMDMs. We now added this term to the Extended Data Figures 1i and l. We additionally performed enrichment analyses on kinase motifs, which revealed the enrichment of the same kinase motifs in both, U937 cells and BMDMs (see below). To demonstrate that the majority of phosphosites detected are not cell type specific, we also stimulated other human cell lines of different tissue origin (HT29, A549, U2OS) and U937 cells with TNF for 15 minutes. Thereby we detected that most of the significantly upregulated phosphorylation events in U937 are also upregulated in other cell lines (see heatmap in response to reviewer 2). Enrichment analysis on regulated phosphorylation events in all of these cell types revealed similar enrichment terms (see below). This confirms that our observations in U937 cells is not restricted to this cell line.

Kinase motif enrichment analysis (Fisher's exact test) of phosphosites significantly upregulated in BMDMs (15 m) and U937 cells (15 m, time course experiment) (p-value < 0.02).

Enrichment analysis (Fisher's exact test) of phosphosites significantly upregulated upon TNF signaling in U937, A549, HT29 and U2OS cells (p-value < 0.02). The enrichment analysis was added to Extended Data Fig. 1.

6. The results of the subcellular localization of the dynamic TNF-mediated phosphorylation events are quite interesting. The data show that many proteins have dual localization in the basal and treated

condition. There are documented examples that phosphosite will regulate the protein subcellular localization. One inherent question: will this localization depend on phosphorylation (site-dependent) ? How to link the proteome-based localization data to infer the potential regulation of localization from individual phosphoprotein of different site ?

These are interesting points. To address the first question: Indeed, our data show that many translocations induced by TNF are dependent on phosphorylation. This can be shown by the inhibition of TAK1, which almost completely abrogates phosphorylation downstream of the TNF signaling receptor. This has a major impact on the translocation of a range of proteins, for example the nuclear translocation of NF κ B1 (Figure 3 d-f). To address the second question: We also attempted to identify phosphorylation events important for protein localization by filtering for phosphosites detected in a fraction where the corresponding protein was missing (Extended Data Figure 3 g-j). This can occur when a small fraction of the protein translocates due to phosphorylation, as the phospho-enrichment strategy facilitates the detection of the phosphopeptide compared to the low abundant unmodified protein.

Though the Figure 3 shows the summary of overall change, the relationship between subcellular localization of the individual proteins at basal level, TNF activation and TAK1 is missing. Presentation of selected examples, such as TNIP1, to show both temporal change and subcellular distribution will be helpful. TNIP1 was detected residing in the cell membrane after stimulation. What is the localization at basal level ? After the TAK1 inhibition, was it reallocated or stay in the basal localization ? Such data will be interesting to show the two dimensional map of phosphorylation-mediated signaling in time and space.

We agree with the reviewer that our data contains much more information than depicted in Figure 3. We stimulated the cells for 15 min with TNF for the fractionation experiment as this time point resulted in the highest number of significantly increased phosphorylations.

TNIP1 is primarily localized in the cytosol (see below). Upon TNF stimulation only a small fraction of the TNIP1 pool in the cell translocates to the membrane. Accordingly, we did not detect a decrease of TNIP1 in the cytosol. In contrast, a large pool of NF κ B1 translocates during TNF stimulation into the nuclear fraction and therefore we were able to detect a reduction of NF κ B1 levels in the cytosol, which does not occur in TAK1 treated cells (see below). Below we also show levels of other important TNF signaling proteins and their distribution across the cellular fractions. Distribution of all detected proteins across the fractions can also be found on our website.

Log2 protein intensities across cytosol, membrane and nucleus upon treatment of TNF alone or in combination with TAK1 inhibitor. Proteins part of the TNF receptor complex, which translocate during TNF stimulation were selected.

7. Compared to previous literature, what are the common findings? Any consistent evidences on the TNF signaling pathways? The Authors may further discuss the advancement of this study.

The comparison of our data to the TNF phosphoproteome study of the Choudhary lab, one of the most recent publications, demonstrated many overlapping phosphorylation events (Wagner et al, 2016, EMBO Journal, PMC5007551) (see the Venn diagram below). Comparisons of phosphosite fold changes of the Wagner et al and our data set revealed correlation of phosphosite regulations (scatter plot below). Importantly, Fisher's exact test in both studies revealed proteins involved in NF- κ B signaling and various toll like receptor signaling pathways as most strongly enriched amongst proteins that are phosphorylated during TNF stimulation. While most of the previous TNF phosphoproteome studies analyzed phosphorylations only at early time points of 5 and 15 minutes or at very late time points (24 hours), we conducted a much more highly resolved time course to unravel phosphorylation kinetics. We were also able to assign these regulated phosphorylation events to an upstream kinase. The study of Krishnan et al (2015, Nat. Communications, PMID: 25849741) looked at IKK β substrates alone, but we analyzed many more kinases which identified several novel targets of IKK ϵ and TBK1 downstream of TNF. Additionally, we are the first to provide spatial information of these phosphorylation events and their corresponding proteins. The abrogation of TNF-mediated phosphorylation had a profound impact on many protein translocations and led us to conclude that these phosphorylation events are also important for protein

translocation. We also provide the phosphoproteome of apoptotic and necroptotic cells and identified a crucial role of CDK12/13 for TNF target gene expression. Overall, we do not only provide an unprecedented resource of the TNF phosphoproteome in a myeloid cell line and a primary myeloid cell type (BMDMs), but also identified a key role of CDK12/13 in cytokine production and cell death induction upon TNF stimulation.

Venn diagram showing the overlap of all detected phosphosites in the study of Wagner et al and our study. Scatter plot of phosphosite fold changes of this study vs Wagner et al (blue dots are phosphorylation events that are significantly changing in our study).

Minor comments:

The Extended Data figure legends need to be presented on the same page as the figures

We added the figure legends to the respective Extended Data figures.

REVIEWER COMMENTS

Reviewer #2 (Remarks to the Author):

In the revised version, the authors provide additional data to address the previously raised concerns. However, my main concern on the limitations of the biological significance of the reported findings still stands.

Major Points:

1. I appreciate that the authors made the effort to include more data concerning the validation of their results also in other cell lines. However, the data provided still fail to show convincing evidence that, indeed, the TNF-regulated phosphorylation sites, identified in U937, are shared between different cellular systems and could therefore guide the identification of novel druggable targets. Importantly, it is impossible to draw any conclusion from the heatmap provided in the rebuttal (page 5) as the list of the phosphorylated peptides detected upon TNF stimulation is missing. The only thing that can be concluded from the heatmap (in the present form) is that the U937 show the strongest changes upon TNF stimulation and that many of the changes identified in U937 are only partial or absent in the other cell lines utilized. Moreover, the authors failed to show which of the specific phosphorylation events are indeed cross-conserved and, therefore, of actual biological significance. This figure needs to be revised and provide the appropriate information to answer this concerns. Simply performing a phospho-proteome analysis upon TNF stimulation in different cellular system without providing a detailed analysis on the main changes and functional regulations observed does not provide convincing evidence on how reproducible and biologically important the observed phosphorylation events are.

2. The authors have now provided a detailed explanation of the methodology employed to generate phosphomimetic and -ablating XIAP mutants as well as the western blot data to assess XIAP knockout and reconstitution efficiency. It is unfortunate, however, that the blots provided are of a low quality especially the blots for actin, the control protein. Despite this, my main concern on the functional relevance of the identified phosphorylation sites remains unanswered as the validation of the phosphorylation events detected on XIAP upon TNF in HT29 also fails to show any biological relevance as no effects were observed when XIAP knockout cells were reconstituted with either phosphorylation-mimicking or phosphorylation-preventing mutants. Therefore, the authors should carefully revise their conclusions in the text and properly justify why the validation of the phosphosites detected on XIAP failed to show any functional differences also in a different cellular systems. Importantly, while I understand that the validation of all the proteins significantly phosphorylated upon TNF treatment will require considerable work and it could be beyond the

scope of the present manuscript (as specified in the rebuttal letter), I still believe that the validation of a few crucial and newly discovered phosphosites would be necessary to prove their potential functional relevance in TNF signaling, especially in light of the new negative results provided by the authors on the validation of the phosphorylation events detected on XIAP in HT29 cells.

Minor point:

A proper discussion about the finding that the combination of transcriptional CDK inhibitors with SM and IDN-6556 results in synergistic cell death induction in U937 and that this is independent of TNF is still missing. The new sentence introduced by the authors “Depending on the sensitivity of different cell lines to TNF-mediated apoptotic and necroptotic stimulation, CDK inhibition exacerbates cell death accordingly” (page 19, line 435) is not fully supported by their results and open to interpretation. This should be revised.

Reviewer #3 (Remarks to the Author):

The Authors have provided either clarification or additional analysis to address all my comments. At its current status, I recommend publication in Nature Communications.

Point by point answers to reviews of “**Phosphoproteome profiling uncovers a key role for CDKs in TNF signaling**” by Maria C Tanzer, Isabell Bludau, Che A Stafford, Veit Hornung and Matthias Mann

We are pleased by the reviewers’ positive assessment of our manuscript and their appreciation of the extensive revision we have performed.

We hope to have now addressed the last concerns of the reviewers by highlighting the supplemental data table, which provides all information about the common and differentially regulated phosphorylation events in response to TNF in the different cell lines. Additionally, the Fisher’s exact test analysis resulted in many of the same enrichment terms between different cell lines, when analyzing their significantly upregulated phosphorylation events upon TNF. This indicates activation of the same pathways.

Finally, we demonstrate that XIAP overexpression or reduction does not impact TNF-induced cell death in HT29 cells, which explains the missing impact of XIAP phosphomutant expression. Therefore, we cannot draw any conclusion about their impact on XIAP functionality.

REVIEWER COMMENTS

Reviewer #2 (Remarks to the Author):

In the revised version, the authors provide additional data to address the previously raised concerns. However, my main concern on the limitations of the biological significance of the reported findings still stands.

Major Points:

1. I appreciate that the authors made the effort to include more data concerning the validation of their results also in other cell lines. However, the data provided still fail to show convincing evidence that, indeed, the TNF-regulated phosphorylation sites, identified in U937, are shared between different cellular systems and could therefore guide the identification of novel druggable targets. Importantly, it is impossible to draw any conclusion from the heatmap provided in the rebuttal (page 5) as the list of the phosphorylated peptides detected upon TNF stimulation is missing. The only thing that can be concluded from the heatmap (in the present form) is that the U937 show the strongest changes upon TNF stimulation and that many of the changes identified in U937 are only partial or absent in the other cell lines utilized. Moreover, the authors failed to show which of the specific phosphorylation events are indeed cross-conserved and, therefore, of actual biological significance. This figure needs to be revised and provide the appropriate information to answer this concerns. Simply performing a phospho-proteome analysis upon TNF stimulation in different cellular system without providing a detailed analysis on the main changes and functional regulations observed does not provide convincing evidence on how reproducible and biologically important the observed phosphorylation events are.

We are in complete agreement with the reviewer of the importance of showing that phosphorylation events upon TNF stimulation occur in U937 cells and other biologically relevant cell lines. We believe that in the revised form, this manuscript will be a valuable addition to the endeavor to discover novel

druggable targets for TNF related diseases. Indeed, the heatmap does not show which phosphorylation events are commonly regulated; we therefore have added this information in **supplemental table 1**. That table contains information on all phosphorylation events detected, including the Log2 fold changes and p-values between control and TNF treatment in the different cell lines. It additionally contains columns that indicate whether the changes within cell lines are significant (FDR < 0.05). This allows the readers to filter phosphorylation events of their favorite cell line for changes with a given p- or q-value and fold change, and provides information on whether these changes are detected in other cell lines.

To compare pathways activated during TNF treatment in these different cell lines, we performed a Fisher exact test on significantly upregulated phosphorylation sites in the different cell lines. This demonstrated that most enrichment terms are shared, which indicates that the same pathways are activated. We previously only included this analysis in response to reviewer 3, but want to highlight it here again.

*Enrichment analysis (Fisher's exact test) of phosphosites significantly upregulated upon TNF signaling in U937, A549, HT29 and U2OS cells (p-value < 0.02). The enrichment analysis was added to **Extended Data Fig. 1h**.*

2. The authors have now provided a detailed explanation of the methodology employed to generate phosphomimetic and -ablating XIAP mutants as well as the western blot data to assess XIAP knockout and reconstitution efficiency. It is unfortunate, however, that the blots provided are of a low quality especially the blots for actin, the control protein. Despite this, my main concern on the functional relevance of the identified phosphorylation sites remains unanswered as the validation of the phosphorylation events detected on XIAP upon TNF in HT29 also fails to show any biological relevance as no effects were observed when XIAP knockout cells were reconstituted with either phosphorylation-mimicking or phosphorylation-preventing mutants. Therefore, the authors should carefully revise their conclusions in the text and properly justify why the validation of the phosphosites detected on XIAP failed to show any functional differences also in a different cellular systems. Importantly, while I understand that the validation of all the proteins significantly phosphorylated upon TNF treatment will require considerable work and it could be beyond the scope of the present manuscript (as specified in the rebuttal letter), I still believe that the validation of a few crucial and newly discovered phosphosites would be necessary to prove their potential functional relevance in TNF signaling,

especially in light of the new negative results provided by the authors on the validation of the phosphorylation events detected on XIAP in HT29 cells.

The main intention of the Western blot was to determine the expression levels of the XIAP variants, which the blots indeed confirm. Importantly, even though the expression levels of phosphomutants were below the endogenous level, they still inhibited TNF-induced cell death, indicating that the mutation had no impact on the function of XIAP to inhibit cell death in U937 cells.

Our experiment with HT29 cells is inconclusive. We apologize that we did not include the comparison of wt HT29 cells with HT29 expressing reduced levels of XIAP (see below). It shows that the reduction of XIAP itself in HT29 cells did not change the sensitivity towards TNF-induced cell death. This is also confirmed by the overexpression of wt XIAP, which did not affect TNF-induced cell death (see previous revision). Therefore, we did not expect a difference for the phosphomutants. This indicates that HT29 are not as dependent on XIAP as U937 cells in terms of sensitivity towards TNF-induced cell death. Hence, we conclude that HT29 cells are not a good system to test XIAP phosphomutants in context of TNF-induced cell death and we think it is better not to include the results in the manuscript. We highlighted in the manuscript that the missing functional impact of XIAP mutants in U937 cells could be very well cell type- and stimulation-specific and that these phosphorylation events could be important in other systems.

Figure 1: Cell death analysis of propidium iodide stained HT29 cell treated as indicated for 24 h (TNF 100 ng/ml, SM/Birinapant 5 μ M).

We identified many regulated phosphorylation events that are known to be required for intact signaling downstream of the TNF receptor and we achieved an unprecedented depth in terms of identified phosphosites. Both points are promising indicators that our dataset contains other novel signaling events important for TNF signaling. However, like the reviewer already acknowledged, it remains for the future to identify the specific biological contexts (e.g. cell type, TNF concentration, combination of other stimulation) where these phosphorylation events have functional roles.

Minor point:

A proper discussion about the finding that the combination of transcriptional CDK inhibitors with SM and IDN-6556 results in synergistic cell death induction in U937 and that this is independent of TNF is still missing. The new sentence introduced by the authors “Depending on the sensitivity of different cell lines to TNF-mediated apoptotic and necroptotic stimulation, CDK inhibition exacerbates cell death

accordingly“ (page 19, line 435) is not fully supported by their results and open to interpretation. This should be revised.

We extended the discussion about CDK inhibitors in combination with Smac-mimetic (SM) and IDN-6556 (highlighted in yellow). However, we believe better not to state that this type of cell death is TNF independent, as SM induces TNF, and cell death upon SM and IDN-6556 treatment has been reported to be TNF dependent (Brumatti et al, PMID: 27194727).

We revised the sentence on page 19 by toning down the statement.

Reviewer #3 (Remarks to the Author):

The Authors have provided either clarification or additional analysis to address all my comments. At its current status, I recommend publication in Nature Communications.